# Divisively normalized neuronal processing of uncertain visual feedback for visuomotor learning

Yuto Makino [1,2], Takuji Hayashi [1] & Daichi Nozaki [1✉]

When encountering a visual error during a reaching movement, the motor system improves the motor command for the subsequent trial. This improvement is impaired by visual error uncertainty, which is considered evidence that the motor system optimally estimates the error. However, how such statistical computation is accomplished remains unclear. Here, we propose an alternative scheme implemented with a divisive normalization (DN): the responses of neuronal elements are normalized by the summed activity of the population. This scheme assumes that when an uncertain visual error is provided by multiple cursors, the motor system processes the error conveyed by each cursor and integrates the information using DN. The DN model reproduced the patterns of learning response to 1-3 cursor errors and the impairment of learning response with visual error uncertainty. This study provides a new perspective on how the motor system updates motor commands according to uncertain visual error information.

[1] Division of Physical and Health Education, Graduate School of Education, The University of Tokyo, Tokyo, Japan. [2] Japan Society for the Promotion of Science, Tokyo, Japan. ✉email: nozaki@p.u-tokyo.ac.jp

Accurate movements are maintained across trials by the ability of the motor system to correct movement according to the difference between the actual and predicted sensory information (i.e., sensory prediction error)[1]. However, sensory information is not always reliable enough because of the noise inherent in the environment and the nervous system[2]. The problem of how the brain updates the motor command based on such imperfect sensory information has been investigated by manipulating the degree of uncertainty in the sensory (primarily visual) feedback for the reaching movements: the visual cursor was blurred[3] or the visual feedback was provided by a cloud of dots (Fig. 1a)[4–6]. These previous studies consistently reported that motor adaptation was impaired when the uncertainty of visual feedback information was increased.

The impairment of motor adaptation was generally explained by the framework of statistical estimation. According to the idea of maximum likelihood estimation (MLE), the motor system optimally estimates the error by combining the observed sensory information and the sensory information predicted by the motor command through the forward model[1,3,7–9]. Thus, as the uncertainty of actual sensory information increases, the motor system relies more on sensory prediction, and the resultant reduction of estimated error leads to impairment of the motor adaptation[3,5] (Fig. 1b). Previous studies have proposed that the computation of MLE is implemented by the probabilistic population coding[10–12] by which the activity levels (or gain) of

neurons depend on the uncertainty of the sensory information. Notably, this scheme assumes the presence of uncertainty of sensory information. For example, when visual feedback is provided by a cloud of dots (Fig. 1a), the uncertainty of the feedback is assumed to be encoded by the statistical dispersion of dots. However, considering that each dot should convey reliable visual error information, such statistical encoding is not trivial. A crucial question is how the uncertainty of visual feedback could emerge from such reliable outputs from a population of dots.

In this study, we tried to take an alternative approach. We assumed that all the dots conveyed the visual error information and that the information was integrated by a divisive normalization mechanism to be used for visuomotor learning. Divisive normalization has been proposed as a canonical computational mechanism in neuronal circuits to integrate the outputs from a population of neurons—the neuronal activity is normalized by the pooled activities of neurons[13,14]. This computation explains a wide range of functions regulating neural responses at an early stage of sensory coding[15–18] and higher-order processes, including attention[19], decision making[20,21], or multisensory integration[22,23]. Our previous study[24] demonstrated that the motor learning response to visual and proprioceptive perturbations can be explained by divisive normalization, implying that the computation based on divisive normalization in the neuronal-circuit level could be reflected in the motor learning behavior.

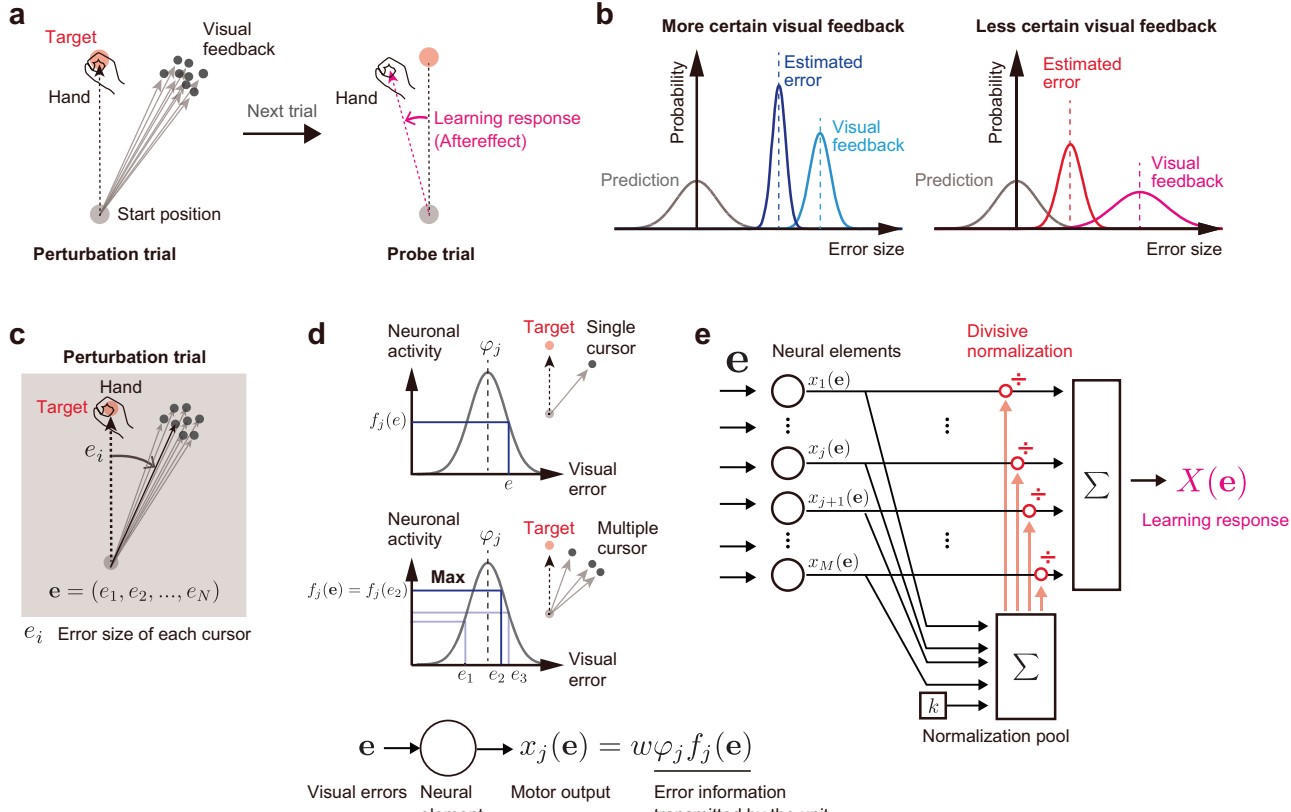

**Fig. 1 The divisive normalization (DN) mechanism processing visual prediction errors. a** In previous studies, uncertain visual feedback was provided by a population of dots (cursors). The learning response to such uncertain visual feedback can be evaluated by measuring the movement direction (i.e., the aftereffect or learning response herein). **b** According to the maximum likelihood estimation scheme, the motor system estimates the error by combining the actual visual information with the visual information predicted by the motor command. The more uncertain the visual information, the smaller the estimated error, which reduces the learning response. **c–e** Proposed model. **c** We consider that each cursor conveys different visual error information. **d** Each neuronal element encodes the visual error according to the Gaussian tuning function with the maximum operation [$\varphi_j f_j(\mathbf{e})$] and transforms the error information to a motor output [$x_j(\mathbf{e})$]. **e** In the DN model, the outputs of the elements are normalized by a DN mechanism before being integrated to produce the learning response.

The present study examined the possibility that the visuomotor learning response to uncertain visual feedback can be explained by the divisive normalization (DN) mechanism. First, we introduced the visuomotor learning model based on DN mechanisms. Then, we validated the model by examining how the reaching movement direction was updated in the next trial (i.e., the aftereffect; we will call it learning response) after single, double, or triple cursors moving in different directions were concurrently presented[25]. Finally, we determined whether the model could reproduce the previously reported empirical result that the learning response decreased as the variance of a cloud of dots increased[4,5]. Additionally, we determined whether the model could predict the recently reported result that the degree of reduction in the learning response with the variance depended on the mean size of visual error[6].

## Results

The present study considered the single-trial-adaptation experiment for reaching movements[24–28], to investigate the learning response. The learning response was often quantified by the changes in reaching movement direction in the next trial after receiving visual error information (learning response in Fig. 1). We first introduce the computational model and then describe the experiments used to validate the model.

**DN model.** To investigate the influence of visual error uncertainty information on visuomotor adaptation, previous studies have used a population of dots (i.e., cursors) as a visual cursor and manipulated the degree of uncertainty by the size of the cursors' distribution (Fig. 1a). In those studies, the visual error information was represented by the mean and variance of the cursors (e.g., Fig. 1b). The current study used a more mechanistic approach: we considered that each cursor conveyed different visual error information ($e_i$) (Fig. 1c), and the information of all cursors was integrated by a DN mechanism[14] to compute the motor command in the next trial (i.e., learning response).

In our model (Fig. 1d, e), a population of neural units encodes multiple visual error information $\mathbf{e} = (e_1, e_2,..., e_N)$ and transforms the information into a compensatory motor command $X(\mathbf{e})$. We assumed that the same units are always recruited as long as the intended movement direction is identical. The tuning function of each unit is a Gaussian function with the maximum operation:

$$f_j(\mathbf{e}) = \max_i \left[ \exp\left\{ -\frac{(e_i - \varphi_j)^2}{2s^2} \right\} \right], \quad (1)$$

where $s$ is the tuning width (22 deg was used in this study following Kang et al.[29]) and $\varphi_j$ is a preferred direction of the j-th unit uniformly distributed from $-180°$ to $180°$ (Fig. 1d). Formally, this should be a circular function (e.g., von Mises probability distribution function). Therefore, it should be interpreted as an approximation that holds only when the $e_i$ and s are relatively small. The max operation in Eq. 1 indicates that the unit responds only to the cursor closest to the peak of its tuning function and ignores the other cursors[13]. Note that this unit transmits the error information as $\varphi_j f_j(\mathbf{e})$ (i.e., the $f_j(\mathbf{e})$ contributes as a gain factor). We assume that each unit generates a compensatory motor command $x_j(\mathbf{e})$ in response to the error information as

$$x_j(\mathbf{e}) = w\varphi_j f_j(\mathbf{e}), \quad (2)$$

where $w$ is a positive constant (Fig. 1d), and the motor commands from all units have been integrated to produce $X(\mathbf{e})$ (In this study, we defined the sign of the aftereffect to be positive for a positive error).

Most importantly, we consider that the motor commands from all units are integrated by a DN mechanism to produce the total compensatory motor command $X(\mathbf{e})$ as:

$$X(\mathbf{e}) = \frac{\sum_{j=1}^{M} x_j(\mathbf{e})}{kM + \sum_{j=1}^{M} x_j^2(\mathbf{e})}, \quad (3)$$

where $k$ is a positive constant (Fig. 1e) and M is the number of units ($M = 3601$ was used in this study, but the results were not affected when $M$ was large enough). The DN mechanism assumes that the outputs from the unit are divided (or normalized) by the summed outputs of units before they are integrated. Thus, unlike the MLE model, the approach by the DN model does not need statistical information such as the mean and variance of cursors. When the DN mechanism is absent (i.e., the second term in the denominator of Eq. 3 is removed) and $\mathbf{e} = e$ (i.e., only one cursor is presented), $X(e)$ can simply be expressed as (see Methods for the details):

$$X(e) = \frac{\sqrt{2\pi}ws}{360k} e, \quad (4)$$

The linear response with the error size has been commonly assumed in previous studies[30–32], indicating that our formalization based on neural units is a natural extension of previous modeling studies. In addition, Eq. 3 shows that the denominator contributes to producing the nonlinear dependence of the learning response on errors. Dividing the denominator of Eq. 3 by $k$ gives

$$1 + \frac{w^2}{kM} \sum_{j=1}^{M} \varphi_j^2 f_j^2(e), \quad (5)$$

which implies that, as long as $M$ is fixed, the term $w^2/k$ represents the strength of nonlinearity (the larger value indicates stronger nonlinearity).

**Experiments used to validate the DN model.** We performed 3 experiments. Experiment 1 aimed to examine whether the DN model can capture the pattern of the learning response when redundant visual error information was imposed by concurrently presenting 1, 2, or 3 visual cursors[25]. Experiments 2 and 3 aimed to examine if the model reproduced the results conventionally explained by the MLE model, i.e., the learning response decreases with the level of the uncertainty of visual error.

**Experiment 1: Learning response to multiple visual errors.** In experiment 1 (8 participants: 5 men and 3 women), we examined a single-trial visuomotor adaptation induced by visual perturbation to the cursor(s). Thirty-nine types of perturbations were used (Fig. 2a). In the single-cursor perturbation condition, the visual error ($e_1$) was imposed by rotating the cursor's movement direction from the target direction ($e_1 = 0°$, $\pm7.5°$, $\pm15°$, $\pm30°$, and $\pm45°$ [9 types]). In the double-cursor perturbation condition, 2 cursors were moved concurrently in different directions. Each cursor has different visual errors ($e_1$ and $e_2$ are a combination of $0°$, $\pm15°$, $\pm30°$, and $\pm45°$ excepting $|e_1| = |e_2|$, i.e., $_7C_2 - 3 = 18$ types). In the triple-cursor perturbation condition, 2 cursors' errors were fixed ($e_2 = 30°$ and $e_3 = 45°$), and the error of the remaining cursor was $e_1 = -45°$, $-30°$, $-15°$, $0°$, $15°$, and $22.5°$ (6 types). There were also 6 symmetric patterns of perturbations in the triple-cursor perturbation condition ($e_1 = 45°$, $30°$, $15°$, $0°$, $-15°$, and $-22.5°$; $e_2 = -30°$; and $e_3 = -45°$). One set of trials consisted of 4 trials: a perturbation trial (1 of 39 types was pseudo-randomly chosen) and a probe trial (Fig. 2b) followed by 2 null trials (Fig. 2c). We confirmed that the learning effect was sufficiently washed out by one probe trial and 2 null trials (Supplementary Fig. 1).

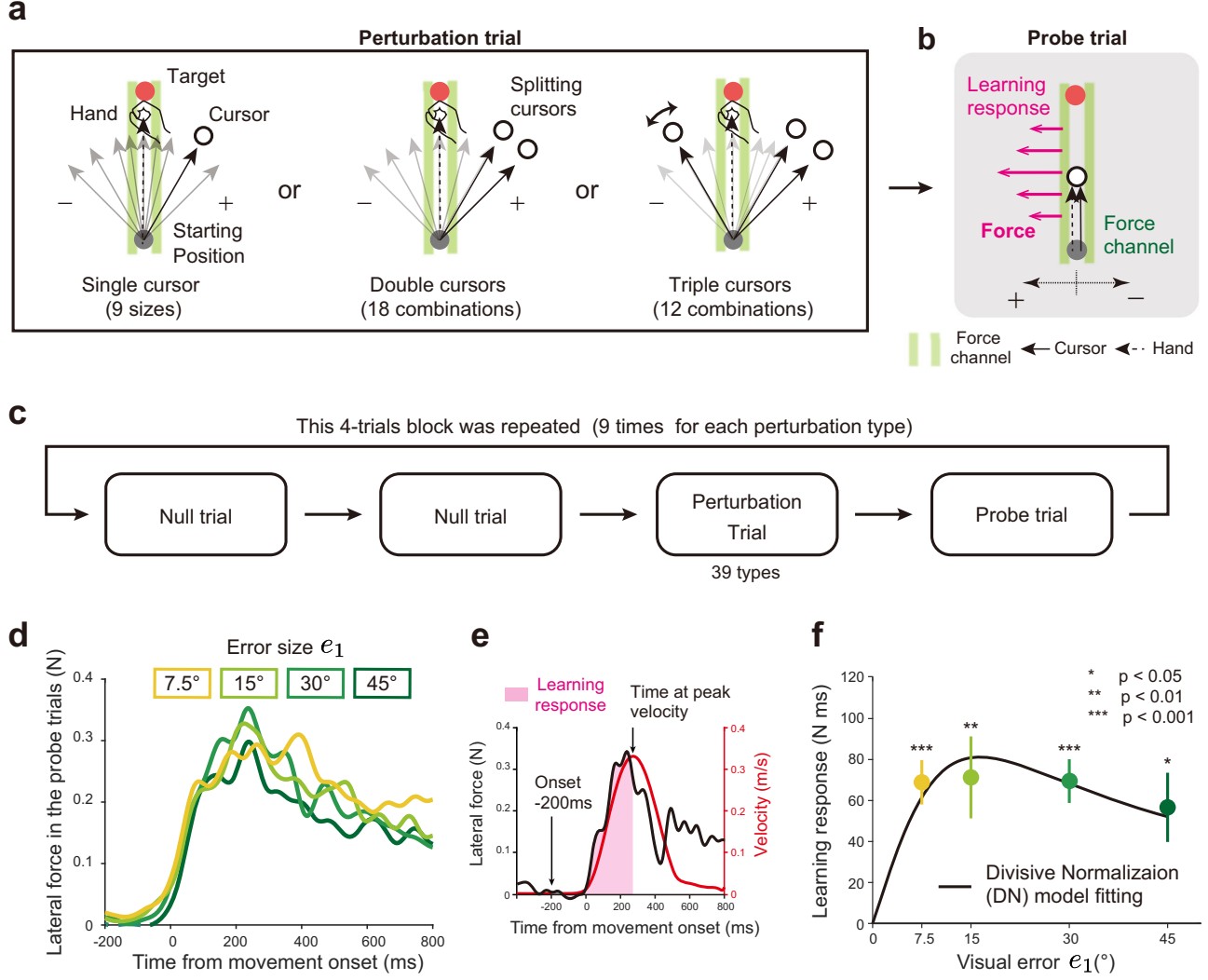

**Fig. 2 Procedures of experiment 1. a, b** Eight participants performed reaching movements towards a front target (distance = 10 cm). In the perturbation trials (**a**), the visual perturbation was imposed on the cursor(s). The number of cursors was 1 (left, single-cursor perturbation condition), 2 (middle, double-cursor perturbation condition), or 3 (right, triple-cursor perturbation condition). In total, there were 39 types of visual perturbations. In the perturbation trial, the hand trajectory was constrained by the force channel. In the subsequent probe trial (**b**), the learning responses were measured by the force exerted against the force channel. **c** Participants repeated a block consisting of 2 null trials, a perturbation trial, and a probe trial. **d** The trace of lateral force exerted against the force channel in the probe trial. **e** The learning response was evaluated as the force integrated from the force onset (200 ms before the movement onset) to the time of peak velocity. **f** The experimental result of the learning response for the single-cursor perturbation condition (green line). The error bars represent the standard error across the participants. The significant learning response was generated in each condition (one sample t-test; $e_1 = 7.5°$, $p < 0.001$; $e_1 = 15°$, $p = 0.009$; $e_1 = 30°$, $p < 0.001$; $e_1 = 45°$, $p = 0.012$). The divisive normalization model (Eq. 6) was fitted with the learning responses (black line).

During the perturbation and the subsequent probe trials, the hand trajectories were constrained in a straight line from the start position to the target using the force-channel method[33]. We measured the lateral force against the force channel to measure the feedback response in the perturbation trial and the learning response in the probe trials.

*Learning response to the single-cursor perturbation condition.* Figure 2d illustrates the evolution of lateral forces against the force channel during the probe trials for the single-cursor perturbation condition. The presence of the learning response was confirmed as the lateral force in the opposite direction to the visual error imposed in the perturbation trials. The learning response was quantified as the sum of lateral forces integrated over the time interval from the force onset to the time at the peak handle velocity (i.e., feedforward component; Fig. 2e).

Note that this measure is a proxy of the change in the movement direction normally used to quantify the learning response[24].

The perturbations induced significant learning responses for the single-cursor perturbation condition (Fig. 2f) ($e_1 = 7.5°$, t[7] = 6.354, $p < 0.001$; $e_1 = 15°$, t[7] = 3.564, $p = 0.009$; $e_1 = 30°$, t[7] = 6.504, $p < 0.001$; $e_1 = 45°$, t[7] = 3.354, $p = 0.012$). However, there were no significant differences between the perturbation sizes (F[3,21] = 0.783, $p = 0.517$), indicating that the learning response did not increase linearly with the size of perturbation but was saturated[6,24–26,34,35]. Notably, this saturation of the learning response with the error size was naturally reproduced by the DN model because of the normalization effect[24]. Indeed, the DN model (Eqs. 1–3) can be well fitted with the learning responses for the single-cursor perturbation condition. In the case of single-cursor perturbation, Eq. 3 can be simplified as (see

Methods for the details):

$$X(e) = \frac{2\sqrt{2\pi}wse}{720k + \sqrt{\pi}w^2s(s^2 + 2e^2)}, \qquad (6)$$

We fit the learning responses $X(e)$ with Eq. 6 to obtain the unknown parameters $w$ and $k$ ($s$ was fixed at 22°). Figure 2f shows that the fitted model was able to reproduce the saturation effect ($R^2 = 0.9542$, $w = 5.3271 \times 10^{-4}$, $k = 7.7806 \times 10^{-7}$). Data for Fig. 2 are available in Supplementary Data.

*Learning response to double- and triple-cursor perturbation conditions.* We used the DN model (Eqs. 1–3) identified by the learning responses of the single-cursor perturbation condition (Fig. 2f) to predict the learning responses for the double- and triple-cursor perturbation conditions. The model predicted that, in the double-cursor perturbation condition, the learning responses did not increase by adding another cursor in the same directions ($e_2 = 15°$, 30°, and 45°) (Fig. 3a). In contrast, a substantial impact was observed in the other perturbation types: the addition of another cursor moving in the target direction ($e_2 = 0°$) reduced the learning responses, and the addition of another cursor in the opposite direction ($e_2 = -15°$, $-30°$, and $-45°$) further decreased or reversed the learning responses (Fig. 3a).

The empirically observed pattern of the learning response for the single- and double-cursor perturbation conditions (Fig. 3b) was in good agreement with the model's prediction (Fig. 3a) ($R^2 = 0.9284$: This was in contrast to the result of the MLE model described later [$R^2 = -0.1221$]). To quantitatively examine the similarity, we also characterized the differences among the perturbation conditions by analyzing the learning responses averaged over $e_1 = 15°$, 30°, and 45° (the inset in Fig. 3c). The model predicted that the averaged responses were not so different between the single and $e_2 = 15°$, 30°, and 45° (Fig. 3c). However, they were considerably reduced for $e_2 = 0°$, $-15°$, $-30°$, and $-45°$ compared with the single-cursor perturbation condition (Fig. 3c). The prediction was consistent with the experimental data (Fig. 3d). Indeed, one-way repeated measures analysis of variance (ANOVA) indicated that the averaged learning responses significantly differed between the size of $e_2$ (F[7,49] = 24.828, $p = 4.512 \times 10^{-14}$); the post hoc test revealed that the learning response was not significantly changed with the addition of another cursor in the same direction ($e_2 = 15°$, 30°, and 45°, $p > 0.05$ by Holm correction), but it significantly decreased with the addition of cursor perturbation in the opposite direction ($e_2 = -15°$, $-30°$, and $-45°$, $p < 0.001$ by Holm correction) or in the 0° direction ($e_2 = 0°$, $p = 0.003$ by Holm correction). There were also significant differences between $e_2 = 0°$ and $e_2 = 15°$ ($p = 0.007$ by Holm correction), between $e_2 = 0°$ and 30° ($p < 0.001$ by Holm correction), between $e_2 = 0°$ and $e_2 = 45°$ ($p = 0.011$ by Holm correction), and between $e_2 = 0°$ and $e_2 = -30°$, $-45°$ ($p < 0.001$ by Holm correction).

Figure 3e shows the DN model prediction of how the learning responses were different between the double-cursor perturbation conditions ($e_2 = 30°$ or $e_2 = 45°$) and the triple-cursor perturbation conditions ($e_2 = 30°$ and $e_3 = 45°$). The model predicted that the concurrent presentation of additional 2 cursors ($e_2 = 30°$ and $e_3 = 45°$) had no significant impact on the learning responses for the double-cursor perturbation conditions (Fig. 3e). In particular, the learning responses should overlap between the triple-cursor condition and the double-cursor condition for $e_2 = 45°$. The experimental results (Fig. 3f) were consistent with this prediction, although the prediction did not perfectly match the experimental data (e.g., the difference between the triple-cursor condition and the double-cursor condition for $e_2 = 30°$ was not observed). Two-way repeated measures ANOVA indicated that there was no

significant difference between the learning responses for the double- and triple-cursor perturbation conditions ($e_1 = 15°$, 0°, $-15°$: F[2,28] = 0.364, $p = 0.701$). Data for Fig. 3 are available in Supplementary Data.

*Can the MLE model explain the pattern of learning responses?.* A natural question is if the previously proposed MLE can explain the learning responses of the double- and triple-cursor perturbation conditions. According to the MLE framework, the motor system estimates the error by integrating the observed visual cursor information (observed error) with the information predicted by the motor command (predicted error) (Fig. 1b). Assuming that the uncertainty of both information is represented by a normal distribution, the optimally estimated learning response $x(e)$ to the observed visual error $e$ should be

$$x(e) = c\frac{\sigma_p^2}{\sigma_p^2 + \sigma^2(e)}e, \qquad (7)$$

where $\sigma_p^2$ is the variance of predicted error, $\sigma^2(e)$ is the variance of observed error, and $c$ is a constant. We assumed that $\sigma^2(e)$ depends on the size of $e$; otherwise, the nonlinear learning response to $e$ (Fig. 2f) cannot be reproduced. We also made a natural assumption that the mean value for the predicted and observed error are 0 and $e$, respectively. Considering the previous studies suggesting that the standard deviation of the signals linearly increases with the mean signal intensity[36,37], the standard deviation $\sigma(e)$ can be represented as follows:

$$\sigma(e) = \sigma_v + k_v|e| \qquad (8)$$

where $\sigma_v$ is the standard deviation of visual error when the error is 0, and $k_v$ is a constant. If the same estimation based on the MLE holds for the multiple cursor condition (when the number of cursor is $N$), the learning response can be expressed as follows:

$$x(\mathbf{e}) = c\left(\frac{1}{\sigma_p^2} + \sum_{i=1}^{N}\frac{1}{\sigma^2(e_i)}\right)^{-1}\left(\sum_{i=1}^{N}\frac{e_i}{\sigma^2(e_i)}\right) \qquad (9)$$

As in the DN model, we obtained the $\sigma_p^2$ and $\sigma^2(e)$ by fitting the learning response for the single-cursor perturbation condition with Eqs. 7 and 8, and then predicted the learning responses for double- and triple-cursor perturbation conditions with Eq. 9.

Equations 7 and 8 could reasonably fit the learning responses for the single cursor condition (Fig. 4a; $R^2 = 0.9983$, $c = 2.963 \times 10^5$, $\sigma_v/\sigma_p = 122.2$, $k_v/\sigma_p = 8.055$). The identified MLE model (Eq. 9) predicted that the learning responses increased and decreased, respectively, with the additional perturbation to the same direction ($e_2 = 15°$, 30°, 45°) and to the opposite direction ($e_2 = -15°$, $-30°$, $-45°$) (Fig. 4b). The MLE model also predicted that the learning responses for the triple-cursor perturbation condition exhibited a peculiar modulation pattern with the size of $e_1$ (Fig. 4c). However, the experimental results (Fig. 3b, f) were inconsistent with these MLE model's predictions.

**Experiment 2: learning response to uncertain visual feedback.** Since the double- and triple-cursor perturbation conditions were peculiar from the ecological viewpoint (a cursor split into 2 or 3 pieces), it is still possible that only the MLE model can explain the learning responses for a more realistic case like when the visual error information is provided by a cloud of dots[3–6]. We performed experiment 2 (16 participants: 11 men and 5 women) and experiment 3 (12 participants: 8 men and 4 women) to examine whether the DN model could explain the learning responses when visual feedback (cursors) was provided by a population of cursors.

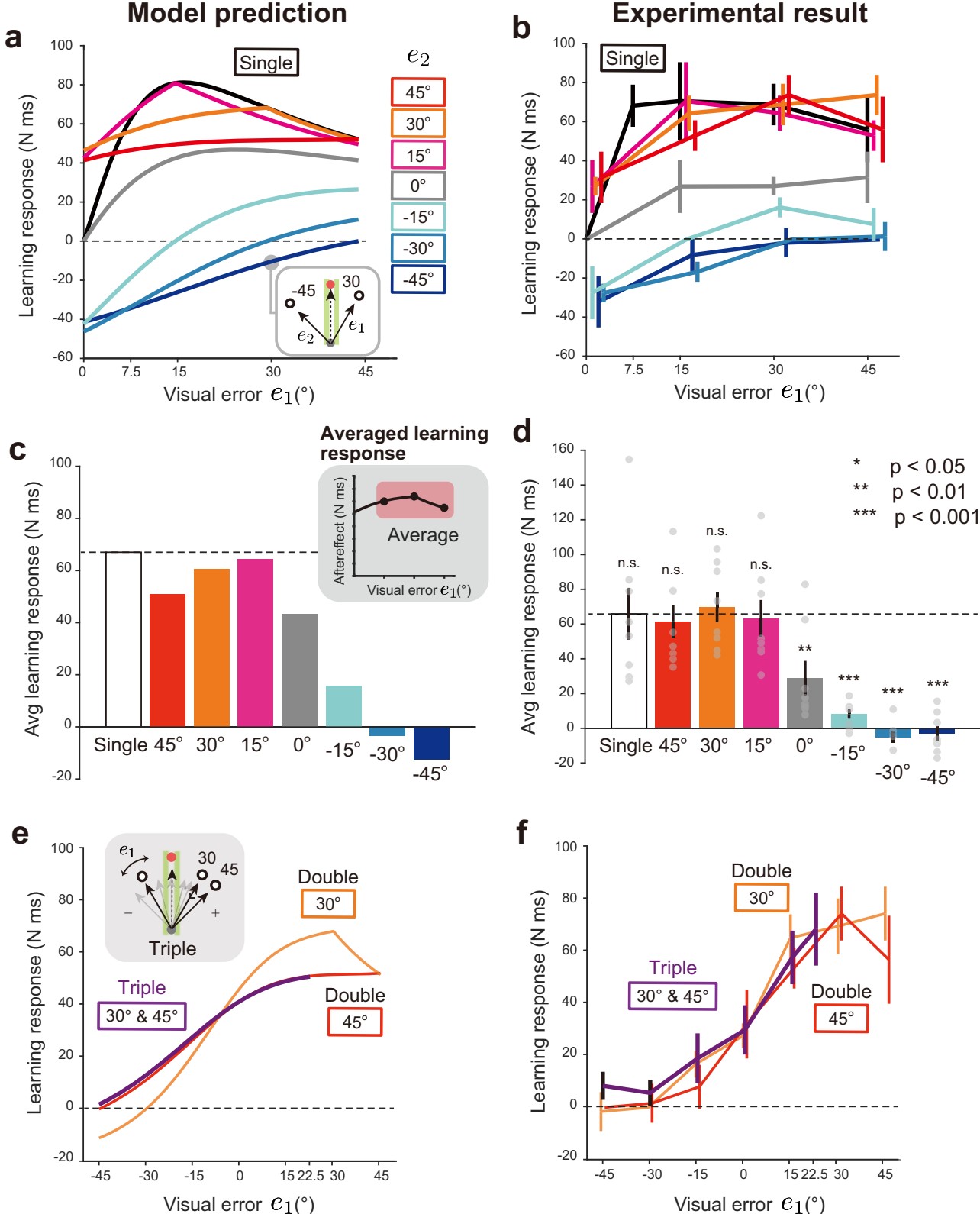

As was in experiment 1, the participants repeated the experimental set consisting of a perturbation trial, a probe trial, and 2 null trials. In the perturbation trials, 5 cursors were concurrently displayed (Fig. 5a). In every perturbation trial, the angles of cursors' movement direction were randomly drawn from 1 of 15 different normal distributions ($N(\mu, \sigma^2)$; $\mu = 0°$, $\pm 14°$, $\pm 40°$; $\sigma = 0°$, $7°$, $20°$) (Fig. 5a). The cursor was visible

throughout the movement in experiment 2 (Fig. 5b), whereas the cursor was only visible after completion of the reaching movement in experiment 3 (Fig. 5c).

First, we predicted the learning responses for these uncertain visual error conditions by the DN model (Eqs. 1–3) whose parameters were identified by fitting the learning responses for the single-cursor perturbation condition (i.e., $\sigma = 0°$) in

**Fig. 3 Learning responses for double- and triple-cursor perturbation conditions in experiment 1. a** The divisive normalization (DN) model prediction for the double-cursor perturbation condition. The parameters of the DN model (Eqs. 1–3) were identified by fitting the learning responses for single perturbations (Fig. 2f). The learning response for the single-cursor perturbation condition is also displayed (black line). **b** The experimentally observed learning responses of the double-cursor perturbation condition in experiment 1 (n = 8). **c** The influence of an additional cursor ($e_2$) on the learning response was evaluated as the averaged learning response for $e_1 = 15°$, $30°$, $45°$ (inset). **d** The experimental results of the averaged learning response in the double-cursor perturbation condition. One-way repeated measures ANOVA revealed that the learning response was significantly changed with the addition of another cursor in the opposite direction ($e_2 = -15°$, $p < 0.001$ ; $e_2 = -30°$, $p < 0.001$; $e_1 = -45°$, $p < 0.001$ by Holm correction), and in the $0°$ direction ($e_2 = 0°$, $p = 0.003$ by Holm correction), but not in the same directions ($p = 1.000$ by Holm correction). **e** The DN model prediction for the triple-cursor perturbation condition compared with that for the double-cursor perturbation condition. **f** The experimentally observed learning responses for the triple-cursor perturbation condition. The error bars represent the standard error across the participants.

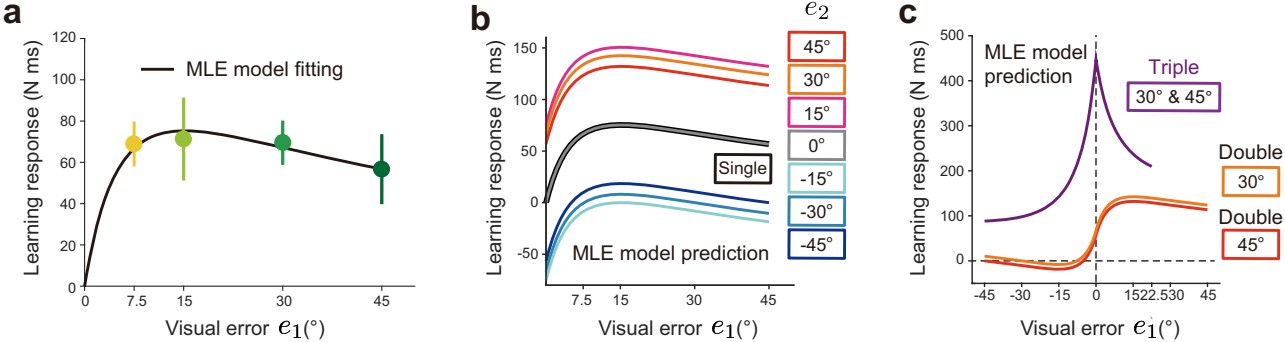

**Fig. 4 The result of model fitting and prediction by the maximum likelihood estimation (MLE) model. a** The modified MLE model (Eqs. 7–8) could reproduce the learning responses for the single-cursor perturbation condition. The error bars represent the standard error across the participants. **b, c** The MLE model (Eq. 9) whose parameters obtained by the learning responses for the single-cursor perturbation condition (Eqs. 7–8) was used to predict the earning responses for double- (**b**) and triple-cursor perturbation conditions (**c**). However, these predictions were totally different from the experimental results (Fig. 3b, f).

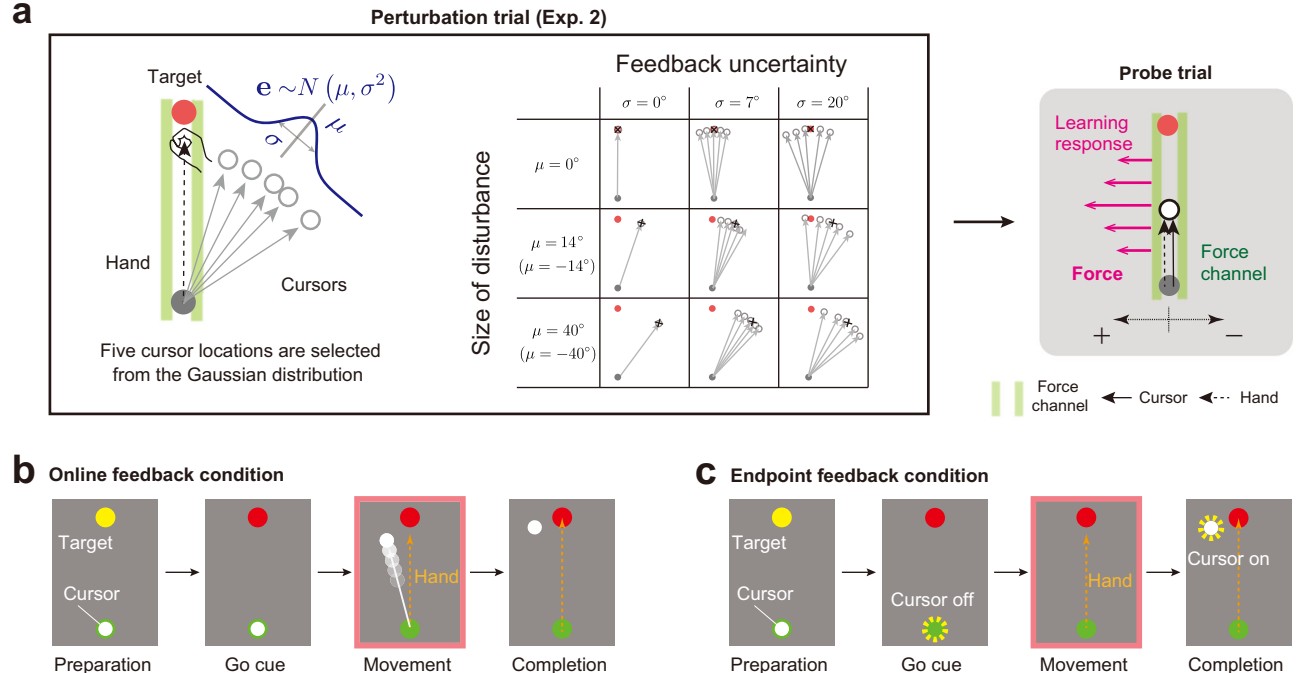

**Fig. 5 Experimental procedures of experiments 2 and 3. a** Participants performed a reaching movement towards a frontal target (movement distance = 15 cm). Five cursors were concurrently displayed as visual feedback in the perturbation trial. The directions of the cursors followed 1 of 15 types of normal distribution $N(\mu, \sigma^2)$ ($\mu = 0°$, $\pm14°$, $\pm40°$; $\sigma = 0°$, $7°$, $20°$). The learning responses were evaluated by the lateral force against the force channel in the next probe trial. **b** In experiment 2 (online feedback condition), the cursors were always visible during the movement. **c** In experiment 3 (endpoint feedback condition), the cursors were visible only after termination of the movement.

## Online feedback condition (Exp.2)

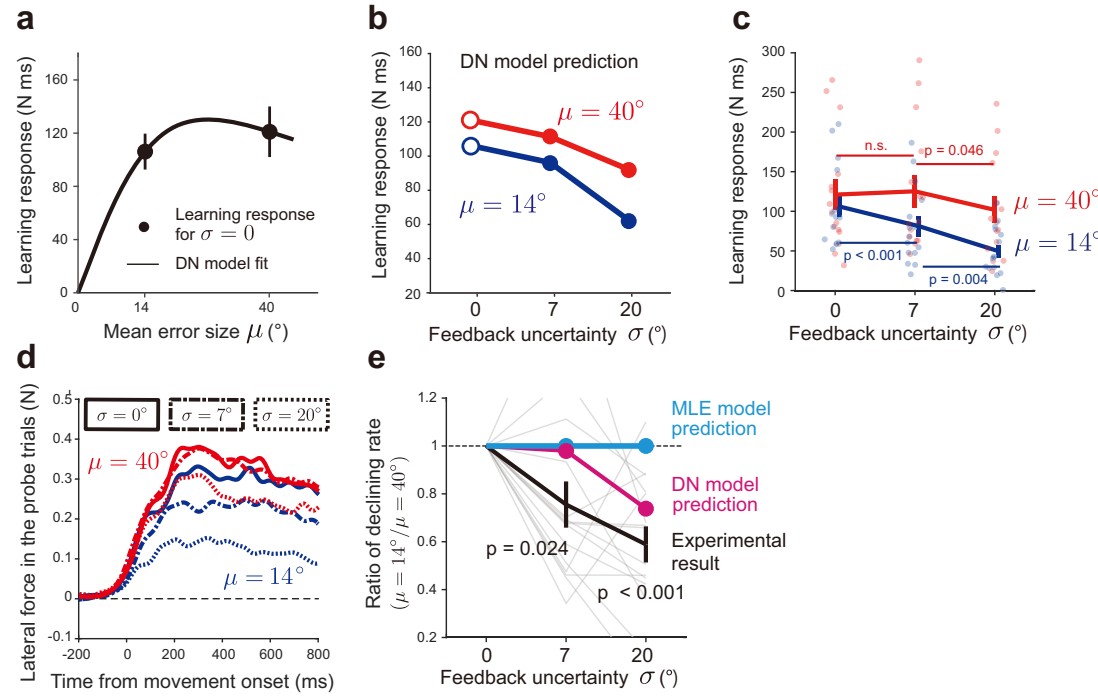

## Endpoint feedback condition (Exp.3)

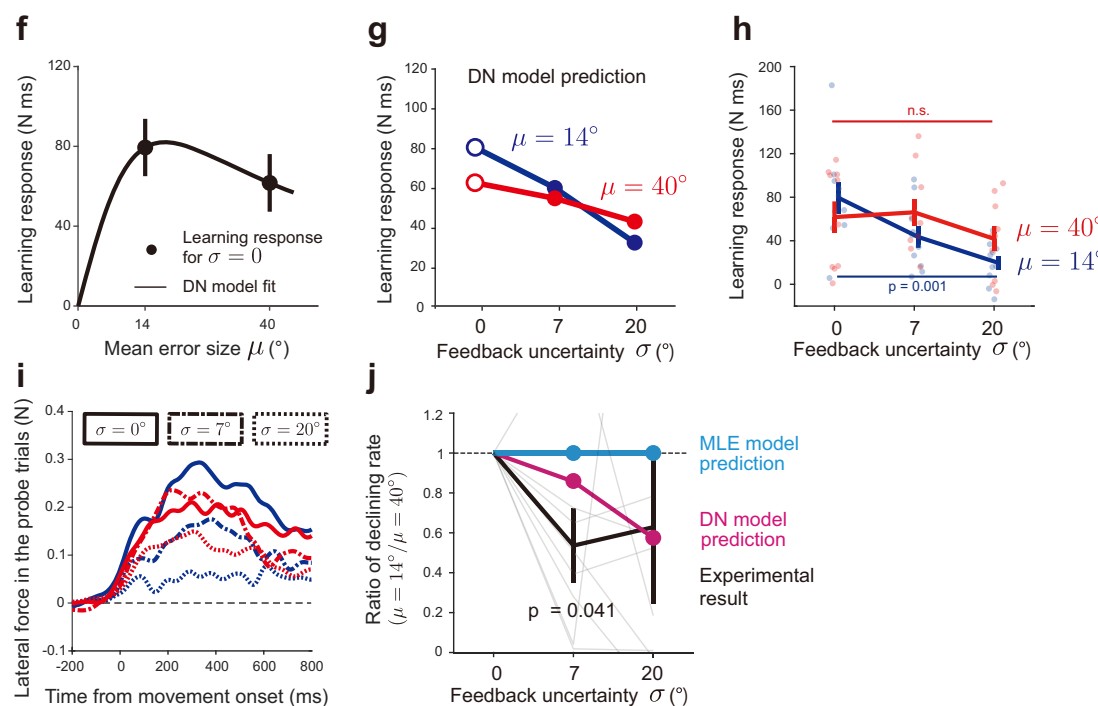

experiment 2 (Fig. 6a) with Eq. 6. As was shown in the results in experiment 1, the fitted DN model captured the modulation of the learning responses with the error size (Fig. 6a; $w = 2.0002 \times 10^{-4}$, $k = 2.1434 \times 10^{-6}$). This identified DN model was used to predict the pattern of the learning responses when uncertainty was introduced to the visual error information (Fig. 6b). When the visual error size was small ($\mu = 14°$), the

increase in uncertainty decreased the learning responses (Fig. 6b), which was consistent with the prediction of the conventional MLE model[3,5].

The observed learning responses in experiment 2 were similar to the pattern predicted by the DN model (Fig. 6c). Two-way repeated measures ANOVA indicated a significant interaction in the learning response between $\mu = 14°$ and $\mu = 40°$ (F[2,28] =

**Fig. 6 The divisive normalization (DN) model predictions and experimental results of experiments 2 and 3. a** The experimentally observed learning responses for no uncertainty conditions (i.e., $\sigma = 0°$) (black circles) in experiment 2 ($n = 15$) and the curve fitted by the DN model (black line). **b** The DN model, whose parameter was obtained by fitting the learning responses for no uncertainty conditions (**a**), was used to predict the change of the learning response to visual feedback uncertainty when the mean error size is small ($\mu = 14°$, blue line) and large ($\mu = 40°$, red line). **c** The experimentally observed learning responses. The significant difference was observed for $\mu = 14°$ conditions between $\sigma = 0°$ and $\sigma = 7°$, $\sigma = 0°$, and $\sigma = 20°$, and between $\sigma = 7°$ and $\sigma = 20°$ (Two-way repeated measure ANOVA; $p = 0.032$, $p < 0.001$, and $p = 0.004$, respectively, by Holm correction), and for $\mu = 14°$ conditions only between $\sigma = 7°$ and $\sigma = 20°$ ($p = 0.046$ by Holm correction). **d** The lateral force traces in the probe trial. **e** The ratio of the declining rate of the learning response with visual feedback uncertainty (the ratio of $\mu = 14°$ to $\mu = 40°$) (experimental result, black; the DN model prediction, magenta; the ordinary MLE model prediction, cyan). The significant difference was observed from the ratio $= 1$ (one sample t-test: $\sigma = 7°$, $p = 0.024$; $\sigma = 20°$, $p < 0.001$). **f–j** the same as panels **a–e** in experiment 3 ($n = 10$). **h** The significant difference was observed for $= 14°$ between $\sigma = 0°$ and $\sigma = 20°$ ($p < 0.001$ by Holm correction), but no simple main effect for $\mu = 40°$ ($p = 0.058$). In panel **j**, the data from 2 participants were excluded because the absolute value of the ratio was extremely large (6 and 20). The significant difference was observed from the ratio $= 1$ (one sample t-test: $\sigma = 7°$, $p = 0.041$). The error bars represent the standard error across the participants.

8.034, $p = 0.002$). There was a significant simple main effect for $\mu = 14°$ between $\sigma = 0°$ and $\sigma = 7°$, $\sigma = 0°$, and $\sigma = 20°$, and between $\sigma = 7°$ and $\sigma = 20°$ ($p = 0.032$, $p < 0.001$, and $p = 0.004$, respectively, by Holm correction). The learning response for $\sigma = 7°$ decreased by $24.82 \pm 8.49\%$ and that for $\sigma = 20°$ decreased by $53.47 \pm 7.44\%$ compared with that for $\sigma = 0°$. In contrast, when $\mu = 40°$, there was a significant simple main effect only between $\sigma = 7°$ and $\sigma = 20°$ ($p = 0.046$ by Holm correction). The learning response for $\sigma = 20°$ decreased by $7.09 \pm 17.52\%$ and that for $\sigma = 7°$ increased by $9.88 \pm 13.72\%$ compared with that for $\sigma = 0°$. These characteristics were also evident for the temporal change of the lateral force against the force channel (Fig. 6d).

Tsay et al.[6] recently reported that when the visual error size ($\mu$) was large, the declining rate of the learning responses with the visual feedback uncertainty ($\sigma$) (relative to the learning response when $\sigma = 0°$) was suppressed. To quantify this effect, we calculated the declining rate of the learning response for $\sigma = 7°$ and $\sigma = 20°$ relative to that for $\sigma = 0°$ for each of visual error size (i.e., $\mu = 14°$ $\mu = 40°$) and the ratio between them (Fig. 6e). Consistent with the finding by Tsay et al.[6], the ratio of the declining rate was significantly smaller than unity for $\sigma = 7°$ ($t[14] = -2.533$, $p = 0.024$) and $\sigma = 20°$ ($t[14] = -5.447$, $p < 0.001$) (the black line in Fig. 6e). Notably the ordinary MLE model (i.e., the model in which the $\sigma(e)$ in Eq. 7 was a constant) predicted that the ratio was always 1 irrespective of the visual feedback uncertainty (the cyan line in Fig. 6e). This was because the term $\sigma_p^2/(\sigma_p^2 + \sigma^2)$ in Eq. 7 was not influenced by the mean size of error ($\mu$). However, the DN model partly reproduced the decrement of the declining rate with the visual feedback uncertainty (the magenta line in Fig. 6e). These DN predictions were not substantially influenced by the number of cursors (Supplementary Fig. 2).

Experiment 3 examined the case in which the visual feedback was visible only after the movement ended (Fig. 5c). The endpoint feedback has been more commonly used in previous studies than the online feedback[3,5,6]. Although the magnitude of the learning response for the endpoint feedback condition was generally smaller than that for the online feedback condition[24], a similar pattern of dependence of learning responses on feedback uncertainty was observed (Fig. 6f-g). Two-way repeated measures ANOVA indicated the presence of a significant interaction in the learning response between $\mu = 14°$ and $\mu = 40°$ ($F[2,18] = 4.188$, $p = 0.032$). There was a significant simple main effect for $\mu = 14°$ between $\sigma = 0°$ and $\sigma = 20°$ ($p < 0.001$ by Holm correction). In contrast, there was no significant simple main effect for $\mu = 40°$ ($F[2,18] = 3.348$, $p = 0.058$). The DN model (Eqs. 1–3), whose parameters were obtained by fitting the learning responses for the single-cursor perturbation condition (Eq. 6) (Fig. 6f; i.e., $\sigma = 0°$) in experiment 3 (Fig. 6g; $w = 4.7525 \times 10^{-4}$, $k = 2.1102 \times 10^{-6}$), could capture the learning response modulation for the endpoint

feedback condition (Fig. 6h). These characteristics were also evident for the temporal change of the lateral force against the force channel (Fig. 6i). Furthermore, the ratio of the declining rate for $\mu = 14°$ to that for $\mu = 40°$ was smaller than unity for $\sigma = 7°$ ($t[7] = -2.495$, $p = 0.041$), which was consistent with the DN prediction (Fig. 6j). Data for Fig. 6 are available in Supplementary Data.

**Pattern of the feedback responses**. In experiments 1 and 2, the visual error information was visible throughout the movement, which induced feedback responses in the perturbation trial. Considering that a feedback response in the perturbation trial could serve as a teaching signal for the learning response in the subsequent trial[28,38], the pattern of learning responses explained by the DN model could already be seen in the pattern of the feedback responses. We examined this possibility by quantifying online feedback responses as the integrated lateral force over the time interval to the time at the peak hand velocity for 100 ms (Fig. 7a).

Figure 7a illustrates how the feedback responses induced by a single visual perturbation in the perturbation trials of experiment 1 depended on the visual error size. As compared with the learning responses (Fig. 2f), the linearity of feedback response modulated with the size of the error was greater. We fit the data with the DN model (Eq. 4) for the single-cursor perturbation condition and obtained the parameters (Fig. 7a; $R^2 = 0.9664$, $w = 4.0541 \times 10^{-4}$, $k = 9.2642 \times 10^{-5}$). The greater linearity was reflected in the smaller value of $w^2/k$ (Eq. 5, feedback response: $w^2/k = 1.7741 \times 10^{-3}$, learning response: $w^2/k = 0.3647$). Using the identified DN model, we predicted the feedback responses for the double- (Fig. 7b) and triple-cursor perturbation conditions (Fig. 7c). These predicted patterns were consistent with the experimentally observed feedback responses (Fig. 7b, c). Moreover, the DN model, fitted with the online responses for the single cursor condition (i.e., $\sigma = 0°$) in experiment 2 ($w = 2.3700 \times 10^{-4}$, $k = 1.2071 \times 10^{-5}$), could reproduce the modulation pattern of the online responses induced by uncertain visual cursors (Fig. 7d). These results suggested that the divisive normalization pattern could also explain the modulation pattern of the online feedback response, but the nonlinearity was weaker for the feedback response than for the learning response. Data for Fig. 7 are available in Supplementary Data.

## Discussion

The motor system maintains accurate movements by constantly updating the motor command according to the movement error. A significant problem is how the motor system estimates movement error in the presence of noise and uncertainty in environments and the central nervous system. Previous studies have proposed the idea that the motor system optimally estimates the error by integrating the actual sensory information and the

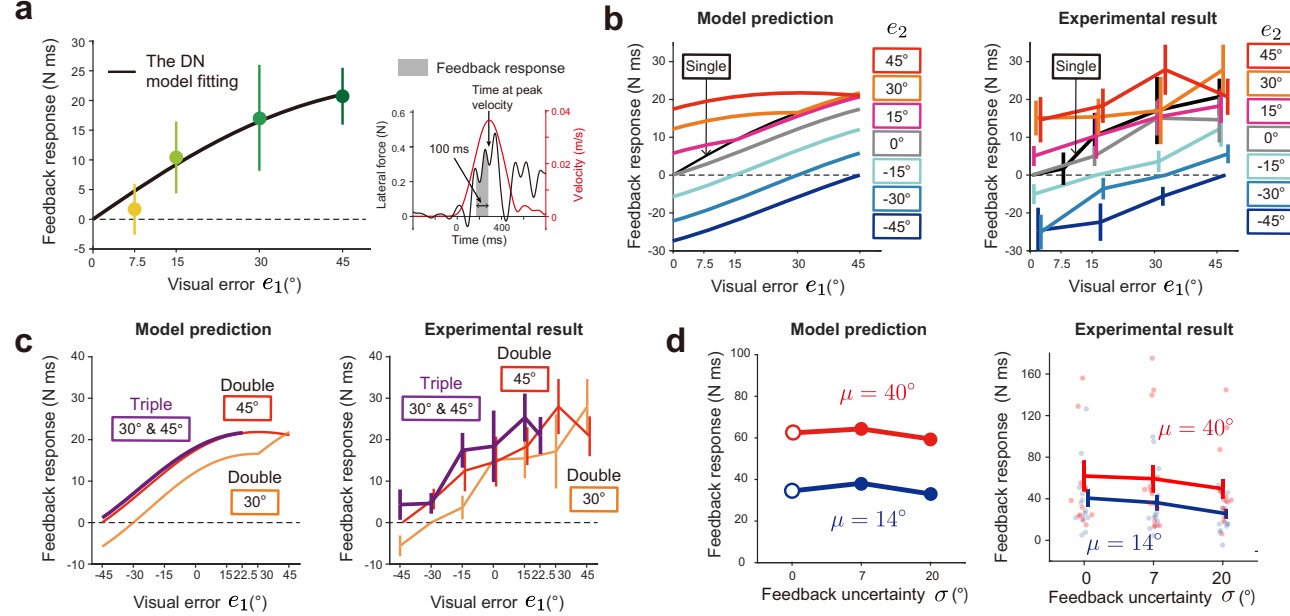

**Fig. 7 The pattern of the feedback response in experiment 1 and 2. a** The experimental result of the feedback responses for the single-cursor perturbation condition in experiment 1 ($n = 8$). The feedback response was quantified by the force impulse from T-100 to T (T: the timing at peak velocity) (right panel). The DN model can be fitted well with these feedback responses (left panel). **b** The DN model, fitted by the feedback responses for single perturbations, to predict the feedback responses for double perturbations (left panel). The experimentally observed feedback responses are consistent with the DN model prediction (right panel). **c** The DN model prediction (left panel) and experimental result of the feedback responses for triple perturbations (right panel). **d** The DN model, whose parameter was obtained by fitting the feedback responses for no uncertainty conditions in experiment. 2, to predict the change of the learning response to visual uncertainty (left panel), and the experimentally observed feedback responses in experiment 2 (right panel). The error bars represent the standard error across the participants.

sensory information predicted by the motor command. This optimal estimation scheme can clearly explain the well-known phenomenon that the learning response (i.e., the aftereffect) is reduced as the uncertainty of visual error feedback increases[3,5].

In contrast to the conventional idea, the present study has proposed a more mechanistic computational scheme. Crucially, we assume that the uncertain visual error information is processed by a DN mechanism[14]. The proposed DN model captured the complicated learning response modulation pattern when the visual error information is provided by a single, double, or triple cursor (Fig. 3). Furthermore, the DN model could reproduce the previous results that the learning response decreases with the level of uncertainty of visual feedback (Fig. 6) without assuming that the uncertainty of visual error information is represented by the mean and variance of the cursors' locations. These results demonstrated that the DN mechanisms contribute to the processing of uncertain and/or redundant visual feedback information in the visuomotor learning system.

The optimal estimation (or MLE) model predicts that the learning response should decrease as the uncertainty of visual error information increases (Eq. 7). However, this model implies that the learning response linearly increases with the visual error size. Thus, additional mechanisms need to be introduced to explain the saturation of the leaning response to the large error[6,24–26,35] (Fig. 2f). Marko et al.[26] assumed that the reduction of the learning response to the larger visual error size reflects the inherent neuronal response in the cerebellum[39]. Wei & Körding[35] proposed that the motor system evaluates the relevance of visual error according to the dissociation between the visual and proprioceptive information. When there is a greater dissociation (i.e., smaller relevance), the motor system relies less on the visual error, leading to the reduction of the learning response. The present study also explored another possibility of the MLE model

by considering that the uncertainty of visual error information increases with the visual error size (i.e., signal-dependent noise[36,37,40] (Eq. 8). In contrast to these ideas, the DN model can naturally reproduce the saturation effect owing to the normalization effect (i.e., the denominator of Eq. 3) without introducing additional mechanisms. Moreover, as discussed in the next section, the aforementioned optimal estimation schemes described cannot reproduce the learning response patterns in the double- and triple-cursor perturbation conditions.

DN provides a mechanism for processing multiple visual feedback information without presuming the statistical property of visual feedback. We tried to validate the mechanism by examining the learning responses in the simplest case in which 2 visual cursors were concurrently provided. The DN model identified by the data of the single-cursor perturbation condition (Fig. 2f) predicted several characteristic patterns (Fig. 3a). When a cursor moving in a direction is accompanied by another cursor moving in the opposite direction or in the 0° direction, the learning response should be reduced. In contrast, when another cursor moves in the same direction, the learning responses are largely unaffected. These predictions were consistent with the experimental result (Fig. 3b). This sub-additivity is a hallmark of the integration of sensory information by divisive normalization[22,23]. We also demonstrated that the experimentally observed patterns are similar to the predicted pattern for the triple-cursor perturbation condition (Fig. 3e, f).

In contrast, according to the idea of the MLE model, the visual error information provided by multiple cursors is optimally integrated under the assumption that the visual information of each cursor has uncertainty. We modified the MLE model by including the signal-dependent noise property (Eq. 8), which is necessary to reproduce the saturation of the learning response in the single-cursor perturbation condition. After fitting the data

with the modified MLE model (Eqs. 7–8; Fig. 4a). We predicted the learning responses for double- and triple-cursor perturbation conditions by Eq. 9 (Fig. 4b, c). The MLE model predicted that the additional cursor changed the learning responses according to the size of additional error (Fig. 4b). The MLE model also predicted that the learning response for the triple-cursor perturbation condition was considerably greater than that for the double-cursor perturbation condition (Fig. 4c). However, changes in the learning response by additional cursor(s) were not observed (Fig. 3b and f). These inconsistencies between the prediction and experimental results suggest that the MLE is unlikely to be a general mechanism to compute the learning response based on the visual error information.

In the double and triple cursor conditions for experiment 1, the participants perceived that the cursor split into 2 or 3 pieces, which was different from the situation when visual feedback was provided with some uncertainties (e.g., a cloud of dots). A crucial problem is whether the DN model can reproduce the well-known phenomenon that the learning response decreases with the level of visual error uncertainty, which was previously explained by the MLE model[3,5]. Using the DN model, we predicted the learning responses when 5 dots whose movement directions were sampled from normal distribution were concurrently displayed. The DN model predicted the decrease in the learning response with the variance when the mean error size was small ($\mu = 14°$) (Fig. 6b, g), and the experimental results validated this prediction (Fig. 6c, h). Thus, the DN model can reproduce the learning response pattern not only for double and triple cursor conditions but also for a noisy cursor condition without any contradiction. Notably, in the DN model's framework, the motor system does not require any a priori information on the mean and variance of the population of dots. It merely encodes the error information conveyed by individual dot and integrates the information. Thus, the DN model provides a mechanistic framework to perform computations similar to the optimal estimation.

Importantly, the DN model provided an additional prediction when the mean size was large ($\mu = 40°$). According to the MLE model, the level of reduction of the learning responses did not depend on the mean size. Yet, the DN model predicted that the level of reduction was smaller for the larger mean size (Fig. 6b, g). Indeed, the experimental results supported the prediction by the DN model (Fig. 6c, h). Recently, similar results were reported when the uncertainty of visual feedback was manipulated[6] or when the adaptation was tested for low vision participants[41]. The authors tried to explain the result by assuming that the distribution of dots is truncated below a certain threshold due to the inherent saturation property of the learning response (i.e., the learning response remains almost unchanged to the error beyond the threshold level). However, it should be noted that their scheme was also based on the MLE model combined with the saturation effect. The DN model is superior in that it can explain the saturation effect and the integration of multiple error information in a unified framework.

In experiments 1 and 2, the visual cursor was visible throughout the movements in the perturbation trials. Although we instructed the participants not to respond explicitly to the visual error, the visual error implicitly elicits the feedback response[42,43]. According to the feedback error learning hypothesis[38], the feedback motor command is used as a teaching signal for the motor command in the subsequent trial. Indeed, a previous study demonstrated that the learning response in the probe trial is similar to the feedback response in the perturbation trial[28].

The experimental result demonstrated that the DN property in the learning response was unlikely to be derived from the DN property in the feedback response. First, the response patterns were considerably different between the learning response and the feedback response: the feedback response pattern exhibited greater linearity as reflected by the smaller $w^2/k$ value (i.e., the normalization effect was weaker, see Eq. 5) (Fig. 3 versus Fig. 7). Thus, even if the learning response is produced by the feedback response as proposed by the feedback error learning hypothesis, the DN mechanism could contribute in the neuronal processes transforming the feedback response to the learning response. Second, even when the feedback response was absent (i.e., experiment 3: the endpoint feedback condition), the learning response still demonstrated the DN property (Fig. 6e–h), clearly demonstrating that the DN property in the learning response could be generated independently of the feedback response. Further studies are necessary to clarify how the motor system uses the online and endpoint visual errors to generate the learning response and how the interaction causes the DN property in the learning response.

The present study firstly demonstrated that the DN mechanism accounts for the motor adaptation to redundant and/or uncertain visual information. Do these behavioral results reflect the way of neuronal processing for visuomotor adaptation? Alvarado et al.[44] recorded the activity of neurons in the superior colliculus responding to visual (unisensory neurons) or both visual and auditory information (multisensory neurons). When 2 visual stimuli are provided concurrently, the authors found that the responses for >60% of neurons are smaller than the summation of responses evoked by individual visual stimulus. Such a sub-additive response is one of the signatures of the DN integration mechanism[14,22,23]. Several other studies have also reported that the DN mechanism successfully accounts for the response of V1 neurons to multiple visual stimuli[16,45,46]. Thus, the DN mechanism is ubiquitous for the neuronal processing of visual information. To our knowledge, no previous studies have investigated if the neurons in the cerebellum, which is responsible for motor learning, have a similar integration mechanism. However, the complex spikes of the Purkinje cell, which is thought to encode the error signal[47], did not increase with the visual error size but decreased as the visual error size became greater[26,39]. Such a nonlinear response could reflect that the activity of Purkinje cells also follows the DN mechanism, which needs to be confirmed in future studies.

In summary, we demonstrated the possibility that the visuomotor learning system processes visual error information by a divisive normalization mechanism. Differently from the conventionally proposed optimal estimation mechanism, the DN mechanism does not assume the statistical property of the uncertainty in the visual feedback (the distribution of cursors). Instead, the DN mechanism assumes that the visual error information conveyed by each cursor is integrated to determine the learning response. This mechanism was able to explain the learning responses when 1, 2, or 3 cursors were concurrently displayed. Furthermore, it reproduced the well-known phenomenon that the learning response decreased with the level of uncertainty in the visual feedback. Considering the explanatory power, the proposed idea provides a novel view of how the motor learning system updates the motor command according to the visual error information.

## Methods

**General experimental procedures**. Thirty-six right-handed participants (24 men and 12 women aged 20–35 years) volunteered for the 3 experiments. Before the experiments started, we explained the experimental procedures, and written informed consent was obtained from all participants. The Office for Life Science Research Ethics and Safety of the University of Tokyo

approved the experiments. All ethical regulations relevant to human research participants were followed.

The participants performed reaching movements with their right hand holding the handle of a manipulandum (KINARM End-point Lab, Kinarm, Kingston, Canada). A horizontal screen was set above the handle to display a start position and a target (5 mm in diameter), and a white cursor (experiments 1 and 2: 5 mm in diameter, experiment 3: 2.5 mm in diameter) indicating the handle position. This screen prevented the participants from directly seeing their arm and handle. To reduce unnecessary movement, the wrist joint was constrained by a brace and the participants' bodies were fixed to a chair by belts.

The green target was displayed at a certain distance (experiment 1: 10 cm; experiments 2 and 3: 15 cm) away from the start position in the straight-ahead direction. A few seconds after the participants held the handle into the start position, the target color turned to magenta, which signaled "go." The participants were asked to move the handle of the KINARM robot toward the target as smoothly and as straight as possible and not to use any explicit strategy even when the cursor was perturbed. The cursor was always visible in experiments 1 and 2, whereas the cursor was visible only after completion of the reaching movement in experiment 3. To maintain a constant velocity across the trials, the warning message ("fast" or "slow") appeared at the end of each trial when the peak velocity of the handle was too fast (>450 mm/s) or too slow (<250 mm/s). The participants maintained the handle at the end position until the robot automatically returned the handle to the start position (~1.5 seconds). Before each experiment, the participants practiced 50 standard reaching movements to achieve stable movements at the appropriate distance and in the range of appropriate velocities.

*Experiment 1: Influence of presenting multiple cursors.* In experiment 1 (8 participants: 5 men and 3 women), we examined a single-trial visuomotor adaptation induced by visual perturbation to the cursor(s). Thirty-nine types of perturbations were used (Fig. 2a). In the single-cursor perturbation condition, the visual error ($e_1$) was imposed by rotating the cursor's movement direction from the target direction ($e_1 = 0°$, $\pm7.5°$, $\pm15°$, $\pm30°$, and $\pm45°$ [9 types]). In the double-cursor perturbation condition, 2 different cursors were moved synchronously in different directions. Each cursor has different visual errors ($e_1$ and $e_2$ are a combination of $0°$, $\pm15°$, $\pm30°$, and $\pm45°$ excepting $|e_1| = |e_2|$ i.e., $_7C_2 - 3 = 18$ types). In the triple-cursor perturbation condition, 2 cursors' errors were fixed ($e_2 = 30°$ and $e_3 = 45°$) and the error of the remaining cursor was $e_1 = -45°$, $-30°$, $-15°$, $0°$, $15°$, and $22.5°$ (6 types). There were also 6 symmetric patterns of perturbations in the triple-cursor perturbation condition ($e_1 = 45°$, $30°$, $15°$, $0°$, $-15°$, and $-22.5°$; $e_2 = -30°$; $e_3 = -45°$).

During the perturbation (Fig. 2a) and the subsequent probe trials (Fig. 2b), the hand trajectories were constrained along the straight line between the start position and the target by using the force channel[33] implemented by a virtual spring (6000 N/m) and dumper (100 N/(m/s)). There were two reasons why we used the force channel instead of measuring the ordinary movement direction. First, this procedure, which combined multiple visual rotations and the force channel, enabled us to completely control the size of the visual error without introducing the proprioceptive error in the perturbation trial (Fig. 2a). Secondly, the force output is more sensitive for the changes in the motor command, as the movement direction is influenced by biomechanical factors such as the inertia and viscosity of the limb.

One set of trials consisted of 4 trials (Fig. 2c): a perturbation trial (1 of 39 perturbation types was pseudo-randomly chosen), probe trial, and 2 null trials without the force channel to wash out the adaptation effects. After 50 practice trials, one cycle was repeated 9 times for 39 perturbation types, for a total of 1404 trials (4 trials/cycle × 9 cycles × 39 perturbations).

*Experiments 2 and 3: Influence of visual error uncertainty.* Experiment 2 (16 participants: 11 men and 5 women) and experiment 3 (12 participants: 8 men and 4 women) were performed to verify whether our DN model could predict the effect of visual feedback uncertainty (a population of cursors) on the learning responses. As was in experiment 1, the participants repeated the experimental set consisting of a perturbation trial, probe trial, and 2 null trials. In the perturbation trials, 5 cursors were concurrently displayed (Fig. 5a). In every perturbation trial, the angles of cursors' movement direction were randomly drawn from 1 of 15 normal distributions ($N(\mu, \sigma^2)$; $\mu = 0°$, $\pm14°$, $\pm40°$; $\sigma = 0°$, $7°$, $20°$) (Fig. 5a). One cycle for 15 distributions was repeated 12 times, resulting in the participants performing a total of 720 trials (4 trials/cycle × 12 cycles × 15 distributions). The cursor was always visible (Fig. 5b) in experiment 2, whereas the cursor was only visible after completion of the reaching movement (Fig. 5c) in experiment 3. Three participants were excluded from the analysis because the learning responses were not clearly observed in the single-cursor perturbation condition (1 participant and 2 participants for experiments 2 and 3, respectively).

*Data analysis.* The position and force data of the handle were sampled at a rate of 1000 Hz and filtered by the 4th-order zero-lag Butterworth filter with a cut-off frequency of 10 Hz. The velocity of the handle was obtained by numerically differentiating the position of the handle. The movement onset was determined as the time when the handle velocity reached 5% of the peak velocity. We quantified the learning response in the probe trials as the lateral force impulse (the summation of lateral force over the time interval from the movement onset – 200 ms to the time at the peak handle velocity) (Fig. 2e). We also quantified the feedback response in the perturbation trials as the sum of the lateral force over the time interval from the time at the peak velocity – 100 ms to the time at the peak handle velocity (Fig. 7a). We integrated the lateral force up to the time at the peak velocity, because we focused on only the implicit component of both responses.

In experiment 1, we collapsed the data of the lateral forces to the perturbations $\pm e_1$ in the single perturbation condition. We also collapsed the data for ($e_1$, $e_2$) and ($-e_1$, $-e_2$) (e.g., ($15°$, $-30°$) and ($-15°$, $30°$)) in the double-cursor perturbation condition and the lateral force data for ($e_1$, $30°$, $45°$) and ($-e_1$, $-30°$, $-45°$) in the triple-cursor perturbation condition. In experiments 2 and 3, we collapsed the data of the lateral forces to the perturbation $\pm \mu$ for each of $\sigma^2$.

*Data exclusion.* We excluded the data from the analysis by the following criteria. The lateral force data was discarded when the peak handle movement velocity for either perturbation or probe trial was outside the included range (<250 mm/s or >450 mm/s for experiment 1, <250 mm/s or >500 mm/s for experiment 2 and 3). We also excluded the data when the participants did not stop the movement clearly after the movement offset. The proportion of the excluded cycle was 9% (experiment 1), 2.8% (experiment 2), and 5.5% (experiment 3).

*Model validation and prediction.* To examine the validity of the DN model, we first fit the learning response data of the single cursor condition (experiment 1) with Eq. 6. The data fitting was performed to the data averaged across participants in this study. Then, using the identified parameters $w$ and $k$, we predicted the possible pattern of learning responses for the double- and triple-cursor perturbation conditions (experiment 1) with the DN model (Eqs. 1–3).

We also fitted the learning responses of the conditions with $\sigma = 0°$ in experiment 2 or 3 by Eq. 6 and then used the identified DN model (Eqs. 1–3) to predict how the learning responses were influenced when 5 cursors were concurrently presented. To examine whether the DN model could also capture the modulation pattern of the online feedback response, we followed the same procedure with the feedback responses in experiments 1 and 2 (Note that the feedback response was not induced in experiment 3, because the cursor was not visible during the movement).

Previous studies have proposed an MLE model to explain the effect of visual information uncertainty on the magnitude of the learning response (Eq. 7). The standard MLE model, however, cannot explain the saturation of learning responses with the visual error size. To reproduce the saturation effect, we assumed that the visual error signal has the characteristic of signal-dependent noise (Eq. 8): the standard deviation of the visual error linearly increases with the visual error size[36,37] (refer to Wei & Körding (2009)[35] for another approach). We fit the learning responses of the single-cursor perturbation condition (experiment 1) with the MLE model and identified the parameters (Eqs. 7–8). Then, we predicted the possible pattern of learning responses for the double- and triple-cursor perturbation conditions using Eq. 9.

*Statistics and reproducibility.* One-sample t-tests were used to test whether the significant learning response for the single cursor condition in experiment 1 was elicited (Fig. 2f) and whether the ratio of the declining rate was different from 1 (Fig. 6e, j). One-way repeated measures ANOVA was used to test the effect of the additional second cursor on average learning responses (Fig. 3d). If the main effect was significant, a post-hoc test was performed to determine which additional cursor caused the significant effect. Two-way repeated measures ANOVA was used to test the effect of visual feedback uncertainty and mean visual error size on the learning response (Fig. 6c, h) and the feedback response. When there was a significant interaction between these factors, a simple main effect of visual feedback uncertainty was evaluated for each mean visual error size. The level of statistical significance was $p < 0.05$.

*Analytical solution of learning response to the single-cursor perturbation.* In this section, we explain how Eqs. 4 and 6 can be obtained from Eqs. 1–3 for the single-cursor perturbation condition (i.e., when **e** is a scalar). We assume that the preferred directions are uniformly distributed from −180° to 180°. The term $\sum_{j=1}^{M} x_j(e)$ in the numerator of Eq. 3 is rewritten as $\sum_{j=1}^{M} w\varphi_j f_j(e)\Delta\varphi/(\Delta\varphi)$, where $\Delta\varphi = 360/M$. Since $\sum_{j=1}^{M} \varphi_j f_j(e)\Delta\varphi$ can be expressed as $\int \varphi \exp\left\{-(e-\varphi)^2/(2s^2)\right\}d\varphi = \sqrt{2\pi}se$, $\sum_{j=1}^{M} x_j(e) = \sqrt{2\pi}Mwse/360$. Similarly, the term $\sum_{j=1}^{M} x_j^2(e)$ in the denominator of Eq. 3 can be expressed as $\sqrt{\pi}sMw^2(s^2/2 + e^2)/360$ using the relation $\int \varphi^2 \exp\left\{-(e-\varphi)^2/s^2\right\}d\varphi = \sqrt{\pi}s(s^2/2 + e^2)$. Replacing $\sum_{j=1}^{M} x_j(e)$ and $\sum_{j=1}^{M} x_j^2(e)$ in Eq. 3 with the obtained expressions gives Eqs. 4 and 6.

**Reporting summary**. Further information on research design is available in the Nature Portfolio Reporting Summary linked to this article.

## Data availability

The source Matlab data that support the findings of this study and the programming codes to analyze the data are available in Zenodo with "https://doi.org/10.5281/zenodo.8343357", and raw experimental data is available from the corresponding author upon reasonable request. Data for the figures is provided in Supplementary Material (Supplementary Data).

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

## Acknowledgements
We thank members of the Nozaki Laboratory for their helpful comments and suggestions, and Asako Munakata for coordinating the experiments. This study was supported by grants from the Japan Society for the Promotion of Science (JSPS) Research Fellowships for Young Scientists (22J15425 to Y.M.), JSPS Research Fellowships (26-2174), Overseas Research Fellowship (29-2601), Uehara Memorial Foundation Postdoctoral Fellowship (201931048 to T.H), and KAKENHI (18H03143, 21H04860, and 22K19736 to D.N.).

## Author contributions
Conceptualization, Methodology, Formal Analysis, and Writing: Y.M., T.H., and D.N.; Investigation, Y.M.; Supervision, D.N.

## Competing interests
The authors declare no competing interests.
