## [Peer Review File · Communications Biology]

Reviewers' comments:

Reviewer #1 (Remarks to the Author):

Makino et al. asked whether divisive normalization mechanisms can unify two motor adaptation phenomena: 1) the effect of multiple visual errors, and 2) the effect of visual uncertainty. Through a series of psychophysical experiments, the authors have elegantly demonstrated how divisive normalization can reproduce several very interesting phenomena: a) the non-linear effect of single cursor adaptation, b) unintuitive results from multiple cursors and visual uncertainty (e.g., adding a larger visual error does not increase learning but paradoxically decreases learning), and c) notable patterns in feedforward and feedback responses. Altogether, I think this is a wonderful and elegant paper, one that would be of wide-interest to the motor learning community. I have no major comments, just a few minor suggestions and questions.

1) How does the divisive normalization model explain visual uncertainty effects observed in individuals with low vision (Tsay et al, 2023: Low Vision Impairs Implicit Sensorimotor Adaptation in Response to Small Errors, But Not Large Errors). Note that in this case, there is always one cursor on the screen, with visual noise present in the visual system. Is the explanation that individuals with low vision see one error, but more variably across trials, hence attenuating adaptation? Or multiple errors that divisively normalize on one trial?

3) Can the authors use divisive normalization to also explain the difference in learning between endpoint and online feedback? Based on Figure 6, endpoint feedback shows less adaptation compared to online feedback. Perhaps in the case of online feedback, participants receive greater samples of errors, whereas, in endpoint feedback, there is only one error sample.

4) Could the authors provide details on the number of trials and repetitions in each experiment? I couldn't find this information in the manuscript.

5) Would the authors be open to sharing their data and code for implementing the DN model? It would certainly benefit future studies intending to reproduce this work.

I really enjoyed reviewing the paper. From my subjective point of view, I think the DN model might be a key notion to understanding how efferent motor commands, proprioception, vision and other senses combine to form one unified kinesthetic percept, one that determines the kinesthetic error for motor adaptation.

Warmly,
JT (Jonathan Tsay)

Reviewer #2 (Remarks to the Author):

Review of "Divisively normalized neuronal processing of uncertain visual feedback for visuomotor learning"

Reviewer: Michael Landy

This paper describes a model for motor learning from rotated feedback (i.e., the typical motor-learning paradigm for reaching) in which a population of units tuned for different values of error are error-direction tuned and interact via normalization to determine the estimated error that then leads

to the accommodating response in the subsequent trial. They study this model by various combinations of one to three possibly perturbed (i.e., rotated) cursors. The perturbation trials and the subsequent test trials all involve a force channel, and the sideways forces of the hand are measured.

While I'm generally in favor of and have used normalization models myself (hell, David Heeger and I share a lab and some funding!), I found the model in this paper to be pretty implausible, which made it hard to get motivated by the rest of the paper. I admit, this is a matter of esthetics as well as plausibility. And, the experiments are certainly novel (multiple cursors going in all sorts of combined directions), but are also pretty unnatural. Typical models of motor learning (well, typical of the Ivry lab and some old work of Shadmehr) suggests that there are multiple learning mechanisms at multiple time scales (somewhat irrelevant here where a single trial of learning is all that is analyzed), but where the early mechanism is more cognitive than automatic/cerebellar. I raise this issue because the multiple-cursor stimulus is so odd that I'd imagine participants could well attend to one or the other cursor across trials, resulting in something closer to switching behavior (high variance across trials) rather than the modeled within-trial competitive normalization.

As I said, I found the multiple cursor setup for Expts. 1 and 2 to be a bit odd. For the cloud of dots used in past work (and for perceptual work from my lab, for that matter), the cloud is meant to be interpreted as a noisy indication of a single event (here, hand feedback), and a larger sigma on cloud dot positions does indeed lead to noisier estimation of cloud position. But, 2 or three dots simply won't be perceived like a cloud, I'd imagine, but rather as two (or three) independent cursors. It's hard to imagine that the higher-level, faster-acting visuo-motor learning system would treat those cases (Expt. 1 vs. Expts. 2&3) identically. Again, this is intellectual esthetics, but it just seems weird to try to cover those two cases with the same mechanism.

Minor point: All intended reaches are straight forward, so that you don't have to deal with the motor field's insistence (which I dispute) that visuo-motor learning doesn't generalize across reach directions. However, if you do agree with that lack of generalization, then your units that encode errors would need to have tuning curves that are centered on an error value as described, but ALSO centered on an intended reach direction. That's a bit of a combinatorial explosion (well, squaring) of the number of needed units. Probably worth clarifying exactly what you have in mind. And, of course, units will also need to deal with reach gain as well as direction. (And, if you were me, you'd also need a 2nd set in exocentric coordinates ;^)

Other things (mostly by line number):

48: leaning -> learning

92: You arbitrarily set the tuning width to 22 deg of reach direction, with justification based on a 22 deg bandwidth for V1 neural tuning for orientation. Really????

95: $\phi_j f_j(e)$: It's probably worth clarifying what you have in mind here. f_j isn't just a tuning curve, it also acts like a gain factor that modulates (multiplies) the peak tuning value to yield (as also modulated by w) a suggested accommodative response to the perturbation.

Eq. 3: This is only vaguely like normalization. Normalization is normally applied to a single neuron's response, divisively normalizing it by the responses of nearby neurons. And yes, Heeger's original formulation did allow for a different power in numerator vs. denominator. But, this equation is really about estimating the accommodative response as the average across all reach-direction-selective neurons, averaging each one's suggested accommodative response (normalized). It's also bizarre (to me) given the $\max()$ function in Equation 1, which seems both biologically odd (normally, in analogy

to V1 cells, multiple inputs are summed, not max'd) and computationally arbitrary. Effectively you have each unit respond only to the cursor closest to the peak of its tuning curve, ignoring the others. Finally, logically Eq. 3 should not be an average, but rather a circular average.

Eq. 4: Why do you bother throwing in a conversion to radians here?

115: "evaluated by w^2/k ": I think you mean something like the strength of the nonlinearity is modulated by that factor, yes? "Evaluated by" doesn't parse.

Figure 2f: You might put a space in or something for "Nms". I tripped over that abbreviation until working out that it was newton-milliseconds. So, maybe "N ms" with a blank.

154: I'm an outsider, but is it typically accepted that 2 washout trials suffices?

164: evolvment -> evolution

169-170: Why did you use a force channel rather than allowing the reach to proceed and measuring the angular error (presumably without cursor feedback)?

199: "was consistent with" compared to what other model? The data and predictions in Figure 3 are in separate plots, making it hard to see how well the model fits. It's kind of okay, but clearly misses quantitatively in several ways. So, some measure of quality of fit (absolute, or relative to MLE or something) would be worthwhile. The "fit" in Fig. 2f is particularly weak (the prediction is an interesting curve, but the data, up to the noise, is basically flat. The stream of tests through 214 barely even is convincing as qualitative proof of the adequacy of the model.

The instantiation of an MLE model that you then put forward looks like independent contributions per dot based on scalar variability (well, Weber's Law) for individual dots that are then averaged perfectly (relative to each dot's uncertainty). This assumes perfect knowledge of uncertainty and perfect averaging (noiseless) and, unlike tons of papers (mine included) in the cue-integration literature, where the likelihood function width is measured in a separate experiment, here you merely fit the model without regard to whether your uncertainty function (which is a Weber's law for size of encoded error, which is not what a visual model would likely do) fits actual visual localization data.

301-307: You are drawing random samples of cursor angles, which means the true average error on a trial is not the same as the nominal means of the conditions (0, 7 or 20 deg). Do you base your predictions on the actual mean in a trial or on the condition's expected mean? The former makes more sense to me.

Fig. 6e: Are you really touting a fit of a model that makes a complex nonlinear prediction to only two points?

377: The value of k from this set is two orders of magnitude bigger than the value of k from the first experiment. That seems unreasonable to me.

382: Clarify what you mean by "the 0 conditions"

Fig. 7: Why did you change the window over which you integrate the sideways force compared to earlier in the paper?

Reviewer #3 (Remarks to the Author):

This paper presents and evaluates a new model for how the motor system deals with visual error uncertainty during motor adaptation. In particular, it proposes that, when there is uncertainty about the visual error due to multiple cursors, the motor system determines a combined response to these errors using divisive normalization. This model is contrasted against prevailing models, which involve maximum likelihood estimation, whereby combined motor responses result from a combination of responses to individual cursors based on the mean and variability associated with each cursor.

This is a very interesting topic, and both the findings and model would be of interest to the field. In particular, the model may provide mechanistic insight on how the motor system deals with visual uncertainty during motor adaptation. The paper is well-written (only a few clarifying questions below) and well-illustrated. I only have a few questions regarding how the model is explained, how it can be interpreted (including potential limitations in the present data), and what its consequences might be for our understanding of motor adaptation. I think these questions can be addressed with minor changes.

1) The Experiment 1 data seem to be clearly in line with the DN instead of the MLE model, especially when a triple-cursor condition is concerned. However, Experiment 2 and 3 don't seem that clear: It is hard to tell how strongly the data support the DN model vs. the MLE one because the two models are not compared directly, and the predictions they make are not that saliently different. For example, a difference between the DN and MLE predictions, illustrated in Figure 6b, is that the MLE-predicted response would drop more steeply when going from a sigma of 7 degrees to a sigma of 20 degrees. The data show that indeed there is a drop in the response, but a drop is predicted by both models – how can we use this to distinguish between them? Direct comparison of how each model fits the data could offer some help. But, maybe more convincingly, since, as this paper discusses, a feature of the MLE model is that the decrement ratio of the learning responses with variance is the same irrespective of the mean value, can this ratio be calculated for the present data and compared to examine whether it significantly shifts with the mean value? As it is now, the data in Figure 6 do illustrate that the DN model can indeed reproduce patterns similar to behavior, but do not seem to rule out the MLE approach.

2) Lines 184-185: how was the model fit? (i) were the subject-averages fit (i.e. the 4 points shown in 2f) or 4 points from each individual? And (ii) was zero included as a point in the fit?

3) As the paper states, it may be that a cursor split into 2- or 3- pieces may be peculiar from an ecological viewpoint. Experiments 2 and 3 are motivated partly by the need to address this issue – however, it doesn't seem a big difference to go from 3 cursors to 5 - if the goal is ecological relevance, why not go further? I am afraid of the possibility that, as the number of cursors increases (eg Tsay 2021 used ~20 for their cloud, according to their figures) the outcomes of the two models might converge. If the goal is to examine more ecologically relevant situations, one could examine how the DN model behaves when you have much larger number of points so collectively they approach the appearance of a blurred cursor (like the ones used in previous work). Would the prediction of the model in this situation be distinct from MLE, or would it converge towards the prediction of traditional MLE models?

4) I am not sure 2 trials would be enough to return adaptation to baseline – instead, the baseline could then be slightly shifting as the experiment progresses. Were there any learning effects, whereby responses would systematically changed depending on whether the probe occurred early vs. late in the experiment?

5) Were forces assessed in baseline (i.e. without a perturbation, or before the perturbation was imposed)? Participants can sometimes exhibit force biases at baseline (e.g. Gibo et al., 2013). While these biases would probably be canceled out when averaging across opposing perturbations, they may still add noise – if there are baseline responses recorded it could be helpful to consider subtracting them from the data.

6) The model assumes each neural element generates a motor output, and then motor outputs are combined using divisive normalization. How would the model outcomes change if the errors themselves were combined using divisive normalization, and the combined error led to motor output?

7) Abstract: given that the topic of the paper is DN, it may be worthwhile to provide a one-sentence description of what DN means. Similarly in the main results – while DN is well defined mathematically (e.g. Equation 3) it would be helpful to have an explanation about what DN does and what distinguishes it from more traditional MLE methods.

8) Minor technical question - There doesn't seem to be something wrong with using the error-clamp during error presentation, but was there a reason for using them instead of a more simple design whereby people just experience the reaching error and then adapt to it - without error-clamp trials either for error presentation or for probe? This would allow just using reaching angles as outcomes instead of force profiles.

9) Line 164: the word "evolvment" may suggest things changing with time, rather than differences across conditions which is what is shown in Figure 2d. Consider rephrasing

Reviewer #1 (Remarks to the Author):

Makino et al. asked whether divisive normalization mechanisms can unify two motor adaptation phenomena: 1) the effect of multiple visual errors, and 2) the effect of visual uncertainty. Through a series of psychophysical experiments, the authors have elegantly demonstrated how divisive normalization can reproduce several very interesting phenomena: a) the non-linear effect of single cursor adaptation, b) unintuitive results from multiple cursors and visual uncertainty (e.g., adding a larger visual error does not increase learning but paradoxically decreases learning), and c) notable patterns in feedforward and feedback responses. Altogether, I think this is a wonderful and elegant paper, one that would be of wide-interest to the motor learning community. I have no major comments, just a few minor suggestions and questions.

We would like to thank the reviewer for their favorable evaluation of our paper. We are also grateful for your helpful comments and suggestions, which we have addressed below. In addition to these changes, we have clarified the data exclusion criteria in the methods section (Lines 667–674). We have also corrected some values in experiment 3 due to the adoption of more consistent data exclusion criteria (Lines 368–373) (This did not significantly affect the results).

1) How does the divisive normalization model explain visual uncertainty effects observed in individuals with low vision (Tsay et al, 2023: Low Vision Impairs Implicit Sensorimotor Adaptation in Response to Small Errors, But Not Large Errors). Note that in this case, there is always one cursor on the screen, with visual noise present in the visual system. Is the explanation that individuals with low vision see one error, but more variably across trials, hence attenuating adaptation? Or multiple errors that divisively normalize on one trial?

Response: Thank you for sharing this interesting point regarding the relevance of our work to the attenuation of motor adaptation in people with low vision. In this study, we used the single-trial adaptation paradigm. Therefore, we cannot say how adaptation effects could accumulate across trials. However, we believe that the latter explanation based on the DN mechanism could account for the visual uncertainty effects observed in individuals with low vision. As shown by the results of experiments 2 and 3, the effect of visual feedback uncertainty on motor adaptation was more apparent when the size of the visual perturbation was small (Fig.6). This is consistent with your observation that the effect of low vision on adaptation was only observed for the small error condition. We have revised the text and have cited the paper that you have shared (Lines 510–512).

3) Can the authors use divisive normalization to also explain the difference in learning between endpoint and online feedback? Based on Figure 6, endpoint feedback shows less adaptation compared to online feedback. Perhaps in the case of online feedback, participants receive greater samples of errors, whereas, in endpoint feedback, there is only one error sample.

Response: Thank you for this insightful suggestion. Indeed, as highlighted, the motor system receives a larger volume of error samples during online feedback than during endpoint feedback. However, according to the divisive normalization framework, a greater amount of error information does not necessarily translate to heightened adaptation. This is because the normalization term also increases as error information increases, as depicted in the denominator of Eq.(3).

Recognizing the significance of differentiating the learning responses between the online and endpoint feedback conditions, we have incorporated a detailed explanation in the results section (Lines 364–366).

4) Could the authors provide details on the number of trials and repetitions in each experiment? I couldn't find this information in the manuscript.

Response: We apologize for the omission. We have now added this information in the revised manuscript (Lines 629–630, Lines 641–643).

5) Would the authors be open to sharing their data and code for implementing the DN model? It would certainly benefit future studies intending to reproduce this work.

Response: We have already shared the data at Zenodo.

<https://zenodo.org/record/7390603#.Y4oUr-zP3Sd>

I really enjoyed reviewing the paper. From my subjective point of view, I think the DN model might be a key notion to understanding how efferent motor commands, proprioception, vision and other senses combine to form one unified kinesthetic percept, one that determines the kinesthetic error for motor adaptation.

We are very glad that you enjoyed reviewing this paper. Thank you very much for your evaluation.

Reviewer #2 (Remarks to the Author):

Review of "Divisively normalized neuronal processing of uncertain visual feedback for visuomotor learning"

Reviewer: Michael Landy

This paper describes a model for motor learning from rotated feedback (i.e., the typical motor-learning paradigm for reaching) in which a population of units tuned for different values of error are error-direction tuned and interact via normalization to determine the estimated error that then leads to the accommodating response in the subsequent trial. They study this model by various combinations of one to three possibly perturbed (i.e., rotated) cursors. The perturbation trials and the subsequent test trials all involve a force channel, and the sideways forces of the hand are measured.

We are grateful for your helpful comments and suggestions, which we have addressed below. In addition to these changes, we have clarified the data exclusion criteria in the methods section (Lines 667–674). We have also corrected some values in experiment 3 due to the adoption of more consistent data exclusion criteria (Lines 368–373) (This did not significantly affect the results).

While I'm generally in favor of and have used normalization models myself (hell, David Heeger and I share a lab and some funding!), I found the model in this paper to be pretty implausible, which made it hard to get motivated by the rest of the paper. I admit, this is a matter of esthetics as well as plausibility. And, the experiments are certainly novel (multiple cursors going in all sorts of combined directions), but are also pretty unnatural. Typical models of motor learning (well, typical of the Ivry lab and some old work of Shadmehr) suggests that there are multiple learning mechanisms at multiple time scales (somewhat irrelevant here where a single trial of learning is all that is analyzed), but where the early mechanism is more cognitive than automatic/cerebellar. I raise this issue because the multiple-cursor stimulus is so odd that I'd imagine participants could well attend to one or the other cursor across trials, resulting in something closer to switching behavior (high variance across trials) rather than the modeled within-trial competitive normalization.

Response: We are honored that you have evaluated our paper and we would like to thank you for the important point you have raised. We are of the understanding that the motor system has the ability to automatically process visual feedback information, even if it seems cognitively unnatural. After reading this comment, we examined the possibility

that the motor system dealt with only one cursor of two cursors in a trial (i.e., switching behavior) rather than integrating them. If this was the case, the distribution of the aftereffects should be bimodal. However, this was not the case. As shown in Figure R1, the distribution of aftereffects (z-score) was unimodal, indicating that switching behavior did not happen.

As I said, I found the multiple cursor setup for Expts. 1 and 2 to be a bit odd. For the cloud of dots used in past work (and for perceptual work from my lab, for that matter), the cloud is meant to be interpreted as a noisy indication of a single event (here, hand feedback), and a larger sigma on cloud dot positions does indeed lead to noisier estimation of cloud position. But, 2 or three dots simply won't be perceived like a cloud, I'd imagine, but rather as two (or three) independent cursors. It's hard to imagine that the higher-level, faster-acting visuo-motor learning system would treat those cases (Expt. 1 vs. Expts. 2&3) identically. Again, this is intellectual esthetics, but it just seems weird to try to cover those two cases with the same mechanism.

Response: Thank you for bringing up the distinction between the two cases: the 2/3 cursors and a cloud of cursors. We recognize that, perceptually, these might be interpreted differently. However, our core assertion is that the motor system treats both cases similarly, and this is the primary focus of our study. The dissociation from perceptual processing can be argued based on the motor command updating process, which operates implicitly and more rapidly. In alignment with this, our findings underscore

that the Divisive Normalization (DN) mechanism can consistently reproduce the results across both scenarios without any contradiction. We have added additional text in the revised manuscript to emphasize these points (Lines 491–493, 500–501).

Minor point: All intended reaches are straight forward, so that you don't have to deal with the motor field's insistence (which I dispute) that visuo-motor learning doesn't generalize across reach directions. However, if you do agree with that lack of generalization, then your units that encode errors would need to have tuning curves that are centered on an error value as described, but ALSO centered on an intended reach direction. That's a bit of a combinatorial explosion (well, squaring) of the number of needed units. Probably worth clarifying exactly what you have in mind. And, of course, units will also need to deal with reach gain as well as direction. (And, if you were me, you'd also need a 2nd set in exocentric coordinates ;^)

Response: Thank you very much for your comment. We are unsure if we have understood your comment correctly, but we have attempted to answer it to the best of our understanding. This paper considered the simplest model in which the same units for generating compensatory motor output are always recruited regardless of the error size and type, because the intended movement direction is always straight forward. We have now described this assumption explicitly in the revised manuscript (Lines 90–91).

Other things (mostly by line number):

48: leaning -> learning

Response: Thank you for pointing this out. We have now corrected this (Line 48).

92: You arbitrarily set the tuning width to 22 deg of reach direction, with justification based on a 22 deg bandwidth for V1 neural tuning for orientation. Really????

Response: Yes, this value was taken from the study by Kang (JNS, 2004). A similar value was also adopted in a study by Tanaka and colleagues (JNP, 2009). Importantly, the prediction was not fundamentally influenced by the value around 20–30 deg (Fig. R2).

Figure R2: Effects of tuning width on the aftereffects for the single and double conditions.

95: $\phi_j f_j(e)$: It's probably worth clarifying what you have in mind here. f_j isn't just a tuning curve, it also acts like a gain factor that modulates (multiplies) the peak tuning value to yield (as also modulated by w) a suggested accommodative response to the perturbation.

Response: Thank you very much for your suggestion. We have now added that $f(e)$ contributes as a gain factor (Line 100).

Eq. 3: This is only vaguely like normalization. Normalization is normally applied to a single neuron's response, divisively normalizing it by the responses of nearby neurons. And yes, Heeger's original formulation did allow for a different power in numerator vs. denominator. But, this equation is really about estimating the accommodative response as the average across all reach-direction-selective neurons, averaging each one's suggested accommodative response (normalized). It's also bizarre (to me) given the $\max()$ function in Equation 1, which seems both biologically odd (normally, in analogy to V1 cells, multiple inputs are summed, not max'd) and computationally arbitrary. Effectively you have each unit respond only to the cursor closest to the peak of its tuning curve, ignoring the others. Finally, logically Eq. 3 should not be an average, but rather a circular average.

Response: Thank you very much for your insightful comment.

- 1) Normalization in Eq.3: We believe that the structure of Eq.3 is consistent with your comment. Indeed, Eq.3 assumes that the response of each unit is normalized (and then integrated). The gain function (Eq.1) guarantees that only nearby units contribute to the normalization process.
- 2) Power in the numerator and denominator: We adopted the current model according to Kouh & Poggio (Neural Computation, 2008). They have argued that the power could be different between the numerator and the denominator. Although we can formulate different models (the denominator is $k + 1/M \sum |x_j|^p$) and obtain the power parameter by the data fitting (the value of p is 1.45), we tried to use the simplest model to avoid increasing the number of parameters.
- 3) Usage of $\max()$ function: We apologize for the confusion. Our model assumes that each unit responds only to the cursor closest to the peak of the tuning curve and ignores the others, as you have suggested. We have revised the manuscript text to make this point clearer (Lines 98–99).
- 4) Circular average in Eq.3: We think that Eq.3 itself does not consist of a problem. However, as you have correctly speculated, Eq.1 is a bit problematic. Formally, in this equation, we should use a circular function (e.g., von Mises distribution function).

The current Eq.1 is just an approximation that holds only when the error is small (e.g., < 45 deg). We have added text in the revised manuscript on this issue (Lines 95–97).

Eq. 4: Why do you bother throwing in a conversion to radians here?

Response: We have aimed to consistently used “degrees” throughout the paper. The $\sqrt{2\pi}$ in Eqs.4 and 6 are not related to radians of the perturbation. Instead, this term has been derived from the mathematical integration of the Gaussian tuning function as derived in the Method section (Lines 700–708).

115: "evaluated by w^2/k ": I think you mean something like the strength of the nonlinearity is modulated by that factor, yes? "Evaluated by" doesn't parse.

Response: Thank you for pointing this out. This sentence has now been modified (Line 122).

Figure 2f: You might put a space in or something for "Nms". I tripped over that abbreviation until working out that it was newton-milliseconds. So, maybe "N ms" with a blank.

Response: We have ensured that this is fixed throughout the manuscript (Fig.2, Fig.3, Fig.4, Fig.6, Fig.7).

154: I'm an outsider, but is it typically accepted that 2 washout trials suffices?

Response: In our previous experiences of performing similar experiments, 2 null trials (+ preceding probe trial) are enough to wash out the learning effect. Figure R3 indicates how the lateral deviation of the hand for the single perturbation conditions changed for

the 1st and 2nd null trials after the probe trials. The lateral deviation almost returned to a baseline level.

164: evolvment -> evolution

Response: Thank you for pointing this out. We have now corrected this (Line 171).

169-170: Why did you use a force channel rather than allowing the reach to proceed and measuring the angular error (presumably without cursor feedback)?

Response: We intended to directly measure the motor command by the force output against the force channel. The force output is more sensitive to the change in the motor command than the movement direction because the movement was influenced by the biomechanical factors including the inertia and viscosity of the arm (in other words, the movement direction was like a low pass filtered motor command). This problem was also pointed out by Albert et al. (J Neurosci, 2016). We have explained the reasons for using the force channel in the revised manuscript (Lines 621-626).

199: "was consistent with" compared to what other model? The data and predictions in Figure 3 are in separate plots, making it hard to see how well the model fits. It's kind of okay, but clearly misses quantitatively in several ways. So, some measure of quality of fit (absolute, or relative to MLE or something) would be worthwhile. The "fit" in Fig. 2f is particularly weak (the prediction is an interesting curve, but the data, up to the noise, is basically flat. The stream of tests through 214 barely even is convincing as qualitative proof of the adequacy of the model.

Response: Thank you very much for your valuable feedback. We have now added the R^2 value and have described that the R^2 value for the MLE was negative (Line 207). We also modified our previous sentence to emphasize that we aimed to quantitatively evaluate the similarity (Lines 207–208).

The instantiation of an MLE model that you then put forward looks like independent contributions per dot based on scalar variability (well, Weber's Law) for individual dots that are then averaged perfectly (relative to each dot's uncertainty). This assumes perfect knowledge of uncertainty and perfect averaging (noiseless) and, unlike tons of papers (mine included) in the cue-integration literature, where the likelihood function width is measured in a separate experiment, here you merely fit the model without regard to whether your uncertainty function(which is a Weber's law for size of encoded error, which is not what a visual model would likely do) fits actual visual localization data.

Response: Thank you very much for your comment. The purpose of data fitting in the experiment with the MLE model was to show that it cannot account for the experimental results. In our study, the variances were obtained by data fitting as parameters, which was a fairly advantageous setting for the MLE model. Despite this, the MLE model failed to replicate the experimental results, indicating the structure MLE model was not adequate to explain the experimental results.

301-307: You are drawing random samples of cursor angles, which means the true average error on a trial is not the same as the nominal means of the conditions (0, 7 or 20 deg). Do you base your predictions on the actual mean in a trial or on the condition's expected mean? The former makes more sense to me.

Response: Yes, as already described in Lines 310–313, we drew random samples of cursor angles (i.e., we adopted the former case).

Fig. 6e: Are you really touting a fit of a model that makes a complex nonlinear prediction to only two points?

Response: Yes. The two points [and (0,0)] were enough to estimate the parameters w and k , because the shape of function Eq.(6) was simple as shown in Fig.2f, 6a, and 6f (i.e., it takes a maximum value).

377: The value of k from this set is two orders of magnitude bigger than the value of k from the first experiment. That seems unreasonable to me.

Response: The difference in magnitude was not caused by the experiment. Instead, this difference reflects the difference between the feedback response and the learning response as described in Lines 392–394.

382: Clarify what you mean by "the 0 conditions"

Response: We apologize for this expression being ambiguous. We meant to say the condition in which the feedback uncertainty was 0 ($\sigma = 0$). We have now clarified this in the revised manuscript (Line 397).

Fig. 7: Why did you change the window over which you integrate the sideways force compared to earlier in the paper?

Response: It was not reasonable to assume that the feedback response started from the movement onset. Thus, to calculate the feedback response, we needed to change the window.

Reviewer #3 (Remarks to the Author):

This paper presents and evaluates a new model for how the motor system deals with visual error uncertainty during motor adaptation. In particular, it proposes that, when there is uncertainty about the visual error due to multiple cursors, the motor system determines a combined response to these errors using divisive normalization. This model is contrasted against prevailing models, which involve maximum likelihood estimation, whereby combined motor responses result from a combination of responses to individual cursors based on the mean and variability associated with each cursor.

This is a very interesting topic, and both the findings and model would be of interest to the field. In particular, the model may provide mechanistic insight on how the motor system deals with visual uncertainty during motor adaptation. The paper is well-written (only a few clarifying questions below) and well-illustrated. I only have a few questions regarding how the model is explained, how it can be interpreted (including potential limitations in the present data), and what its consequences might be for our understanding of motor adaptation. I think these questions can be addressed with minor changes.

Response: Thank you very much for your favorable evaluation of our study and for your helpful comments. We have revised the manuscript according to the comments and suggestions, as described below. In addition to these changes, we have clarified the data exclusion criteria in the methods section (Lines 667–674). We have also corrected some values in experiment 3 due to the adoption of more consistent data exclusion criteria (Lines 368–373) (This did not significantly affect the results).

1) The Experiment 1 data seem to be clearly in line with the DN instead of the MLE model, especially when a triple-cursor condition is concerned. However, Experiment 2 and 3 don't seem that clear: It is hard to tell how strongly the data support the DN model vs. the MLE one because the two models are not compared directly, and the predictions they make are not that saliently different. For example, a difference between the DN and MLE predictions, illustrated in Figure 6b, is that the MLE-predicted response would drop more steeply when going from a sigma of 7 degrees to a sigma of 20 degrees. The data show that indeed there is a drop in the response, but a drop is predicted by both models – how can we use this to distinguish between them? Direct comparison of how each model fits the data could offer some help. But, maybe more convincingly, since, as this paper discusses, a feature of the MLE model is that the decrement ratio of the learning

responses with variance is the same irrespective of the mean value, can this ratio be calculated for the present data and compared to examine whether it significantly shifts with the mean value? As it is now, the data in Figure 6 do illustrate that the DN model can indeed reproduce patterns similar to behavior, but do not seem to rule out the MLE approach.

Response: Thank you very much for pointing this out. The main purpose of performing experiments 2 and 3 was to investigate whether the DN model could also predict the decrease in the learning response with visual feedback uncertainty that was previously explained by the MLE model.

However, the attenuation of the degree of learning for larger error mean size was also an important feature that only the DN model could predict. We agree with you on the point that Fig. 6b and f (broken lines) in the original manuscript were not appropriate to clearly show this effect. In accordance with your suggestion, we have calculated the ratio of the declining rate of the learning responses between small ($\mu = 14^\circ$) and large mean error size ($\mu = 40^\circ$). The data shows that, when the visual feedback uncertainty was introduced, the ratio was significantly smaller than 1, which was predicted only by the DN model (the ordinary MLE model predicted the ratio was always 1). We have now revised the manuscript by adding new panels to highlight these features (Fig. 6e and j), and have revised the figure legend and texts accordingly (Lines 350–360, 375–377).

2) Lines 184-185: how was the model fit? (i) were the subject-averages fit (i.e. the 4 points shown in 2f) or 4 points from each individual? And (ii) was zero included as a point in the fit?

Response: (i) The learning responses averaged across participants were used for the data fit. We have described this procedure in the Method section (Lines 680–681).

(ii) No, the learning response to 0 deg perturbation was not included for the fitting. We subtracted the learning response for 0 deg perturbation from the learning response to the other perturbations for each participant. Accordingly, the learning response to 0 deg perturbation was always 0. Thus, substituting the value (0, 0) into the theoretical learning responses (Eq.6) did not influence the data fitting.

3) As the paper states, it may be that a cursor split into 2- or 3- pieces may be peculiar from an ecological viewpoint. Experiments 2 and 3 are motivated partly by the need to address this issue – however, it doesn't seem a big difference to go from 3 cursors to 5 - if the goal is ecological relevance, why not go further? I am afraid of the possibility that,

as the number of cursors increases (eg Tsay 2021 used ~20 for their cloud, according to their figures) the outcomes of the two models might converge. If the goal is to examine more ecologically relevant situations, one could examine how the DN model behaves when you have much larger number of points so collectively they approach the appearance of a blurred cursor (like the ones used in previous work). Would the prediction of the model in this situation be distinct from MLE, or would it converge towards the prediction of traditional MLE models?

Response: Thank you very much for raising this important point. Since we used a relatively large cursor (diameter of 5mm in experiments 1 and 2), we limited the number of cursors to 5 to avoid the overlap of cursors with this size. We agree with you that we should examine what the DN model would predict if the number of cursors increased. Figure R4 indicates the DN prediction when the number of cursors was increased from 5 to 10 and 100. The fundamental pattern of learning response remains unchanged (a further increase in the number of cursors did not change the pattern). We have included this figure in the supplementary material and described the influence when increasing the number of cursors in the revised manuscript (Lines 359–360).

4) I am not sure 2 trials would be enough to return adaptation to baseline – instead, the baseline could then be slightly shifting as the experiment progresses. Were there any

learning effects, whereby responses would systematically changed depending on whether the probe occurred early vs. late in the experiment?

Response: In our experiences of similar experiments, 2 null trials (+ preceding probe trial) are enough to wash out the learning effect. Figure R5 indicates how the lateral deviation of the hand for the single perturbation conditions changed for the 1st and 2nd null trials after probe trials. The lateral deviation almost converged to a baseline level after the 2nd null trials and we expected that the learning effect should be washed out more in the next perturbation trial.

To see the difference in the learning responses between the early and late phases of the experiment, we plotted the learning responses (all perturbation conditions) in the late phase (last 4 cycles) against those in the early phase (first 4 cycles) (Fig.R6). There was no systematic difference between these two phases.

5) Were forces assessed in baseline (i.e. without a perturbation, or before the perturbation was imposed)? Participants can sometimes exhibit force biases at baseline (e.g. Gibo et al., 2013). While these biases would probably be canceled out when averaging across opposing perturbations, they may still add noise – if there are baseline responses recorded it could be helpful to consider subtracting them from the data.

Response: As you have speculated, the effects of such biases (if any) could be canceled out because the data of opposing perturbation patterns were averaged (i.e., $\{f(e)-f(-e)\}/2$). In fact, the subtraction of baseline force did not influence the data, because $f(e)-f(-e)=\{f(e)-f(0)\}-\{f(-e)-f(0)\}$. The data shown in the response above (Fig.R6) also indicates that drifting effect was limited.

6) The model assumes each neural element generates a motor output, and then motor outputs are combined using divisive normalization. How would the model outcomes change if the errors themselves were combined using divisive normalization, and the combined error led to motor output?

Response: Our previous study by Hayashi et al. (J Neurosci, 2020) used a simple model in which the visual and proprioceptive errors were integrated according to the DN model. We could use a similar model: $X(e) = w \sum e_j / (\text{divisive term})$. However, this model cannot properly deal with the case when $e_j=0$. For example, it cannot discriminate the 2 cases: $e=(30 \text{ deg}, 0 \text{ deg})$ (i.e., double cursor condition) and $e=30 \text{ deg}$ (i.e., single cursor condition). Thus, the approach using the neural unit (represented by Eqs.1-3) is necessary.

In this study, we interpreted that $x_j(e) = w \phi_j f_j(e)$ (Eq.2) is the motor output. However, $\phi_j f_j(e)$ can be alternatively interpreted as the error information encoded by the unit. Thus, the model outcomes are not substantially influenced even when this interpretation is adopted.

7) Abstract: given that the topic of the paper is DN, it may be worthwhile to provide a one-sentence description of what DN means. Similarly in the main results – while DN is well defined mathematically (e.g. Equation 3) it would be helpful to have an explanation about what DN does and what distinguishes it from more traditional MLE methods.

Response: Thank you very much for your suggestion. We have added a brief explanation

of divisive normalization in the abstract. We have also added an explanation in the main text (Line 109–112).

8) Minor technical question - There doesn't seem to be something wrong with using the error-clamp during error presentation, but was there a reason for using them instead of a more simple design whereby people just experience the reaching error and then adapt to it - without error-clamp trials either for error presentation or for probe? This would allow just using reaching angles as outcomes instead of force profiles.

Response: We intended to directly measure the compensatory motor command by the force output against the force channel. The force output should be more sensitive to the change in the motor command than the movement direction, because the movement was influenced by the biomechanical factors including the inertia and viscosity of the arm (in other words, the movement direction was like a low pass filtered motor command. This problem was also pointed out by Albert et al. (J Neurosci, 2016). We have explained the reasons for using the force channel in the revised manuscript (Lines 621–626).

9) Line 164: the word “evolvment” may suggest things changing with time, rather than differences across conditions which is what is shown in Figure 2d. Consider rephrasing

Response: As per the suggestion by Reviewer #2, we have changed this to “evolution” (Line 171).

Reviewers' comments:

Reviewer #1 (Remarks to the Author):

The authors have thoroughly addressed all my comments related to the manuscript. While the authors have uploaded their data, it's not clear whether the authors have uploaded code related to how they implemented the model in an accessible/annotated manner. If not, I encourage the authors to do so in aiding replication/reproducibility.

JT

Reviewer #2 (Remarks to the Author):

Re-review of "Dvisively normalized neuronal processing of uncertain visual feedback for visuomotor learning"

Reviewer: Michael Landy

As I said for the first round, and is still true, I am not happy with the form of this "normalization" model, but some of those reasons are simply because the task and the model feel a bit implausible and clunky. Nevertheless, the model is described clearly and does a better job of accounting for the data than some other models, so the result is a reasonable paper. The model requires units that are tuned for different rotational errors, but that only respond to one of the cursors that fall in the unit's receptive field (responding only to the best-tuned stimulus, rather than the usual linear combination). That's pretty odd.

Otherwise, it's kind of a vanilla normalization model. However, the typical normalization model would normalize each of those units separately, leading to a vector of responses (one per normalized, tuned unit). The description (and Fig. 1e) doesn't talk about normalized tuned responses, but only about their sum. In the model and figure, the units are summed and then normalized, whereas the typical approach would be to normalize the individual units (possibly using the same denominator for all, but not necessarily) and THEN a readout would be used for any further processing. If that readout is simply a sum, then normalize-then-pool gives the same output as their pool-then-normalize. This non-traditional description of a normalization model might trip up readers who know the normalization literature, so you might want to clarify this further to put it in typical normalization terms. The $1/M$ in Eq. 3 is also non-standard for that literature, and again it doesn't matter, since it can be absorbed into k .

* Line 113: "When the denominator ... is absent" may throw off a reader, since it's never absent. Maybe you mean in the case where the sum is small and k dominates (i.e., in the non-normalized weak-signal regime). But, if that's what you mean, then the $1/k$ needs to be in Eq. 4. Otherwise, you are just providing a specialization of the numerator, without saying why you are doing this. Note: the math in 700-708 to support this solves for the sum of the $x()$'s, which is needed for Eq. 3, but also solves for the sum of the $x^2()$'s, which isn't needed for that. It it used for Eq. 6, but I get a different answer than Eq. 6 (I could obviously be wrong): you should get k/M , not k , in the denominator (or redefine k) and the 2nd denominator term should, I think, have $\sqrt{2 \pi}$ and should not have the extra s .

* 142 refers to an inset in the figure that isn't there.

* 229: the influence was small in the model, but completely missing in the data (Fig. 3e and 3f really

don't look all that similar, but this is not acknowledged as a model failure).

Reviewer #3 (Remarks to the Author):

Thank you for your thoughtful and thorough responses. I find the additional analyses and clarifications very helpful. I only have two minor points:

1) Regarding having only two washout trials: Figure R3/R5 shows that the learning effect is *not* washed out after two trials (on average, goes down from a 3 to 2 degrees). However, it is helpful to know there are no systematic differences between early and late learning (Figure R6), so I don't think incomplete washout would be an issue. My only suggestion is to briefly comment on that, as readers (especially within the field) would probably wonder why are there only 2 washout trials.

2) Thank you for the analysis of the ratio of declining rate, I think it helps summarize Figure 6. I suggest that you add a short explanation as to how the ratio is calculated, probably in the methods – for example, I would think the black curve in Figure 6e would be the ratio of the blue slope over the red slope from Figure 6c, (maybe after some normalization) but that doesn't seem to be the case.

We would like to thank the reviewers for invaluable comments and suggestions. Below we provide point-by-point responses to the comments. In this revision, we have also added the section of "Statistics and reproducibility section" (Lines 695-797), "Data availability section" (lines 699-702), and the figure captions describing the statistical method and the number of participants (Figs. 2, 3, 6).

Reviewer #1:

The authors have thoroughly addressed all my comments related to the manuscript. While the authors have uploaded their data, it's not clear whether the authors have uploaded code related to how they implemented the model in an accessible/annotated manner. If not, I encourage the authors to do so in aiding replication/reproducibility.

We appreciate your time and effort in reviewing our paper. We will upload the data and the necessary program code before the paper is published.

Reviewer #2:

Re-review of "Divisively normalized neuronal processing of uncertain visual feedback for visuomotor learning"

Reviewer: Michael Landy

As I said for the first round, and is still true, I am not happy with the form of this "normalization" model, but some of those reasons are simply because the task and the model feel a bit implausible and clunky. Nevertheless, the model is described clearly and does a better job of accounting for the data than some other models, so the result is a reasonable paper. The model requires units that are tuned for different rotational errors, but that only respond to one of the cursors that fall in the unit's receptive field (responding only to the best-tuned stimulus, rather than the usual linear combination). That's pretty odd.

We would like to thank the reviewer who, despite having reservations about the model, provided a fair assessment of the paper. Regarding the response of the element, if linear summation is allowed, the output would diverge as the number of stimuli increases. The max function is one of the simplest assumptions to avoid such divergence (a soft-max function is another choice as discussed in Kouh & Poggio (2008)).

Otherwise, it's kind of a vanilla normalization model. However, the typical normalization model would normalize each of those units separately, leading to a vector of responses (one per normalized, tuned unit). The description (and Fig. 1e) doesn't talk about normalized tuned

responses, but only about their sum. In the model and figure, the units are summed and then normalized, whereas the typical approach would be to normalize the individual units (possibly using the same denominator for all, but not necessarily) and THEN a readout would be used for any further processing. If that readout is simply a sum, then normalize-then-pool gives the same output as their pool-then-normalize. This non-traditional description of a normalization model might trip up readers who know the normalization literature, so you might want to clarify this further to put it in typical normalization terms. The 1/M in Eq. 3 is also non-standard for that literature, and again it doesn't matter, since it can be absorbed into k.

We agree with the reviewer on that the model structure shown in Fig.1e is not standard. We have modified Fig.1e (and the caption of Fig.1) so that the sum of the outputs is taken after each output is normalized.

As for the 1/M in Eq.3, the current expression is not standard (and a bit ugly). We have decided to change the expression slightly (Eq.3 and Eq.5). Thanks for these useful comments.

* Line 113: "When the denominator ... is absent" may throw off a reader, since it's never absent. Maybe you mean in the case where the sum is small and k dominates (i.e., in the non-normalized weak-signal regime). But, if that's what you mean, then the 1/k needs to be in Eq. 4. Otherwise, you are just providing a specialization of the numerator, without saying why you are doing this. Note: the math in 700-708 to support this solves for the sum of the x()'s, which is needed for Eq. 3, but also solves for the sum of the x^2()'s, which isn't needed for that. It is used for Eq. 6, but I get a different answer than Eq. 6 (I could obviously be wrong): you should get k/M, not k, $\sqrt{2\pi s^2}$ in the denominator (or redefine k) and the 2nd denominator term should, I think, have $\sqrt{2\pi}$ and should not have the extra s.

As the reviewer correctly pointed out, the expression "the denominator was absent" was strange. We have changed the expression (Lines 102-103) and Eq.4. Regarding the math in the final section of Methods, one of the equations lacked the 's': We have corrected the equation as $\sqrt{\pi} s M w^2 (\frac{s^2}{2} + e^2) / 360$ (Line 689-693). This correction did not influence Eq. 6. We greatly appreciate your thorough check on our paper.

* 142 refers to an inset in the figure that isn't there.

We have removed the word '(inset)' (the caption of Figure 2).

* 229: the influence was small in the model, but completely missing in the data (Fig. 3e and 3f really don't look all that similar, but this is not acknowledged as a model failure).

We have mentioned the discrepancy between the prediction and experimental results (Lines 210-214).

Reviewer #3 (Remarks to the Author):

Thank you for your thoughtful and thorough responses. I find the additional analyses and clarifications very helpful. I only have two minor points:

1) Regarding having only two washout trials: Figure R3/R5 shows that the learning effect is *not* washed out after two trials (on average, goes down from a 3 to 2 degrees). However, it is helpful to know there are no systematic differences between early and late learning (Figure R6), so I don't think incomplete washout would be an issue. My only suggestion is to briefly comment on that, as readers (especially within the field) would probably wonder why are there only 2 washout trials.

Thank you very much for pointing out this important issue. We have added a brief description of this issue (Lines 149-150) and added a figure showing the evidence that 2 washout trials (and one probe trial) sufficiently reduced the learning effect (Supplementary Figure 1).

2) Thank you for the analysis of the ratio of declining rate, I think it helps summarize Figure 6. I suggest that you add a short explanation as to how the ratio is calculated, probably in the methods – for example, I would think the black curve in Figure 6e would be the ratio of the blue slope over the red slope from Figure 6c, (maybe after some normalization) but that doesn't seem to be the case.

We have added a description of how the data in Figure 6e were obtained (Lines 347-349) and slightly modified the following sentence (Lines 349-351).

REVIEWERS' COMMENTS:

Reviewer #2 (Remarks to the Author):

Re-re-review of "Divisively normalized neuronal processing of uncertain visual feedback for visuomotor learning"

Reviewer: Michael Landy

The minor changes are mostly an improvement, but some bugs arrived as a result. Some things are newly noticed.

Eq. 2: Silly point: the response to an error is typically to adjust the reach in the OPPOSITE direction, yet I don't see a minus sign in Eq. 2 or later to make this happen.

Eq. 4: Lost the essential factor of "e"

Line 109: Now you want to divide by kM , not just by k

line 111: Now w^2/kM represents the strength of nonlinearity

Re-re-review of "Divisively normalized neuronal processing of uncertain visual feedback for visuomotor learning"

Reviewer: Michael Landy

The minor changes are mostly an improvement, but some bugs arrived as a result. Some things are newly noticed.

Eq. 2: Silly point: the response to an error is typically to adjust the reach in the OPPOSITE direction, yet I don't see a minus sign in Eq. 2 or later to make this happen.

Thank you very much for this important point. We have added the definition that the aftereffect to a positive error was positive (Lines 79-80, Page 5)

Eq. 4: Lost the essential factor of "e"

We are sorry for this careless mistake. We have corrected Eq.(4) (page 5).

Line 109: Now you want to divide by kM , not just by k

Yes. We could have defined kM as k , but would like to show explicitly that this term is proportional to the number of units (M). This definition is useful to express Eq.(6) without M (i.e., we can explicitly demonstrate that $X(e)$ is not dependent on the number of units).

line 111: Now w^2/kM represents the strength of nonlinearity

As long as the value of M is fixed, w^2/k can be considered to represent the strength of nonlinearity. We have added this description (Line 97 page 5).